# Dramatic differences in carbon dioxide adsorption and initial steps of reduction between silver and copper

Yifan Ye[1,2,3,9], Hao Yang [4,9], Jin Qian[4,9], Hongyang Su[2,5], Kyung-Jae Lee[2,6], Tao Cheng [4,7], Hai Xiao[4,7], Junko Yano [1,8], William A. Goddard III [4,7] & Ethan J. Crumlin [2,3]

Converting carbon dioxide ($CO_2$) into liquid fuels and synthesis gas is a world-wide priority. But there is no experimental information on the initial atomic level events for $CO_2$ electro-reduction on the metal catalysts to provide the basis for developing improved catalysts. Here we combine ambient pressure X-ray photoelectron spectroscopy with quantum mechanics to examine the processes as Ag is exposed to $CO_2$ both alone and in the presence of $H_2O$ at 298 K. We find that $CO_2$ reacts with surface O on Ag to form a chemisorbed species (O = $CO_2^{\delta-}$). Adding $H_2O$ and $CO_2$ then leads to up to four water attaching on O = $CO_2^{\delta-}$ and two water attaching on chemisorbed (b-)$CO_2$. On Ag we find a much more favorable mechanism involving the O = $CO_2^{\delta-}$ compared to that involving b-$CO_2$ on Cu. Each metal surface modifies the gas-catalyst interactions, providing a basis for tuning $CO_2$ adsorption behavior to facilitate selective product formations.

[1] Joint Center for Artificial Photosynthesis, Lawrence Berkeley National Laboratory, Berkeley, CA 94720, USA. [2] Advanced Light Source, Lawrence Berkeley National Laboratory, Berkeley, CA 94720, USA. [3] Chemical Sciences Division, Lawrence Berkeley National Laboratory, Berkeley, CA 94720, USA. [4] Materials and Process Simulation Center, California Institute of Technology, Pasadena, CA 91125, USA. [5] Hefei National Laboratory for Physical Sciences at the Microscale, University of Science and Technology of China, Hefei, Anhui 230026, China. [6] Department of Physics and Photon Science, Gwangju Institute of Science and Technology (GIST), Gwangju 500-712, South Korea. [7] Joint Center for Artificial Photosynthesis, California Institute of Technology, Pasadena, CA 91125, USA. [8] Molecular Biophysics and Integrated Bioimaging Division, Lawrence Berkeley National Laboratory, Berkeley, CA 94720, USA. [9] These authors contributed equally: Yifan Ye, Hao Yang, Jin Qian. Correspondence and requests for materials should be addressed to J.Y. (email: jyano@lbl.gov) or to W.A.G. III (email: wag@caltech.edu) or to E.J.C. (email: ejcrumlin@lbl.gov)

In order to close the anthropogenic carbon circle while creating a sustainable hydrocarbon-based energy cycle, it is essential to discover new electrocatalysts that can efficiently convert carbon dioxide ($CO_2$) into liquid fuels and feedstock chemicals[1–7]. However, $CO_2$ is highly inert, making the $CO_2$ reduction reaction ($CO_2RR$) unfavorable thermodynamically. High throughput experimental and computational screening methods have been employed to discover new advanced $CO_2$ reduction catalysts but these approaches are based on preconceived notions of the reaction mechanisms and have not produced dramatic successes[8–11]. To accelerate progress we believe that it is essential to develop a complete atomistic understanding of how $CO_2$ interacts with and is transformed by the catalyst surfaces to provide guidance to design the catalyst to selectively tune the mechanisms for adsorption and activation.

Electrocatalysts such as Au, Ag, Zn, Pd, and Ga are known to yield mixtures of CO and $H_2$ at varying ratios depending on the applied voltage[12–16], with Ag exhibiting particularly high activity and selectivity to CO vs. $H_2$. We sought to obtain a comprehensive understanding of how $CO_2$ and $H_2O$ molecules adsorb on the Ag surface and interact to initiate $CO_2$ dissociation and subsequent product formation.

Here we report our findings combining in-situ ambient pressure X-ray photoelectron spectroscopy (APXPS) with quantum mechanics (QM), which leads to unexpected and exciting findings for $CO_2$ surface adsorption on Ag surface that are quite different than observed previously for Cu surfaces. We find that physisorbed linear ($l$-) and chemisorbed bent ($b$-) $CO_2$ are not stable on pure Ag (111) surface, but rather gaseous ($g$-) $CO_2$ reacts with on-top surface oxygen (O) atoms on Ag to form a chemisorbed species ($O=CO_2^{\delta-}$). This surface species stabilizes up to four adsorbed $H_2O$, through forming hydrogen bonds (HBs). We also find that a pair of surface $H_2O$ stabilize $b$-$CO_2$ on the Ag by forming two HBs between the $H_2O_{ads}$ and $CO_2$. The QM results and experimental observations suggest that the ($O=CO_2^{\delta-}$)-$(H_2O)_n$ clusters are the main surface adsorbates with $CO_2$ and $H_2O$ co-adsorption. On Ag we find a very different and more favorable mechanism of activating $CO_2$, involving the $O=CO_2^{\delta-}$, compared to that involving $b$-$CO_2$ on Cu. Ag and Cu surfaces differ in both the chemical speciation and the respective adsorption energies, operating entirely differently for the first step of activating $CO_2$.

## Results

**Dramatic differences in $CO_2$ adsorption between Ag and Cu.** For both Ag and Cu surfaces, we find that oxygen plays an essential role to induce reactions involving $CO_2$ and $H_2O$, but the consequences for each metal are dramatically different. The stability of surface and subsurface O in Ag and Cu surfaces are compared (see Supplementary Note 1; Supplementary Figs. 1 and 2; Supplementary Table 1), where we find that subsurface O, which stabilized both the $l$- and $b$- $CO_2$ in the Cu system (Fig. 1a, b)[17,18], is not stable on Ag; quantum mechanics (QM) finds that putting an O in an Ag subsurface site goes without a barrier to an on-top three-fold ($\eta_3$) site (Supplementary Fig. 2). These on-top surface O atoms interact with gaseous ($g$-) $CO_2$ to form a chemisorbed surface carbonic acid-like species in which two O on the C bind to adjacent Ag bridging sites, while the third O forms a C double bond ($C=O$) perpendicular to the surface. We denote this carbonic acid like adsorbate as $O=CO_2^{\delta-}$ to indicate that the negative charge is on the two O binding to the Ag surface. Our combined QM and experimental results show that only $O=CO_2^{\delta-}$ is stable prior to exposure to $H_2O$. For Cu only unreactive $l$-$CO_2$ is stable without $H_2O$.

Adding $H_2O$ to the surface with $O=CO_2^{\delta-}$ and $g$-$CO_2$ leads to two kinds of structures stable at 298 K and the applied pressures, carbonic acid-like species attaching up to four water, ($O=CO_2^{\delta-}$)-$(H_2O)_{1-4}$, and $b$-$CO_2$ attaching two water. The observation of the surface cluster of ($O=CO_2^{\delta-}$)-$(H_2O)_{1-4}$ is different from the previous understanding of $CO_2$ on metal surfaces, which all involve ($b$-$CO_2$)-$(H_2O)_n$ configurations.

**$CO_2$ adsorption on Ag surfaces.** The (111) surface is closest packed, making it the most favorable facet for Ag and Cu. Indeed experimental evidence shows that silver (and Cu) at high temperature exposes this facets[17,19,20]. Thus our simulations compare results on the Ag (111) surface with experimental observations on vacuum annealed polycrystalline Ag surface.

We started by carrying out QM studies to examine the stability of various surface adsorbates on pristine Ag surfaces, considering both $l$- and $b$- $CO_2$. The optimized structure for $l$- and $b$- $CO_2$ is found to be unfavorable with $E_{ads}$ (QM electronic binding energies) $= -0.15$ eV and $\Delta G = +0.19$ eV, and $E_{ads} = +0.77$ eV and $\Delta G = +1.13$ eV, respectively (Supplementary Fig. 2). These and all other $\Delta G$ values are from QM calculations including zero point energy, entropy, and specific heat to obtain $\Delta G$ at 298 K and at the pressure quoted.

**$CO_2$ adsorption on oxygen treated Ag surfaces.** We started the calculation by considering the possible promotion effect of sublayer oxygen that we found previously to stabilize $CO_2$ adsorption on Cu surface. However, for Ag the QM finds that subsurface O minimizes to the O at the surface. In the presence of isolated surface O, we found that $l$-$CO_2$ has $\Delta E_{ads} = -0.21$ eV, but $\Delta G = +0.13$ eV (Supplementary Note 2; Supplementary Fig. 2). Thus a pressure of ~30 Torr would be required to stabilize $l$-$CO_2$ on the O/Ag surface at 298 K. This contrasts with observations for Cu, where subsurface O stabilized the adsorption of $l$-$CO_2$ on Cu surface under 0.7 Torr $CO_2$ partial pressure at 298 K[17] (Fig. 1b). This attraction resulted from the subsurface O in a tetrahedral site inducing $Cu^+$ character into the single Cu atom above it on the surface, which stabilized the $l$-$CO_2$. This oxygen promotion effect is not observed for Ag because the O is chemisorbed on top of the Ag, which does not facilitate Ag oxidation (to $Ag^+$)[19,21–25]. This contrasting result provides fresh insight into the tunability of $CO_2$ adsorption on metal surfaces.

We evaluated the stabilization of $b$-$CO_2$ next to surface $O_{ad}$ on Ag, but the QM minimizes to form a surface carbonic acid-like species (Supplementary Fig. 2) with a $C=O_{up}$ double bond (1.222 Å) pointing up while the other two O bind to adjacent three fold Ag (111) sites with C-O lengths of 1.365 Å and 1.354 Å and O-Ag distances of 2.276 Å (Fig. 2a). This is not an ionic carbonate possessing three similar O atoms, as had been speculated previously[26–28]. The $CO_2$ bonding energy to form surface $O=CO_2^{\delta-}$ is $\Delta E_{ads} = -0.75$ eV, $\Delta G = -0.28$ eV. We denote this carbonic acid-like adsorbate as $O=CO_2^{\delta-}$ to indicate that the negative charge is on the two O binding to the Ag surface. The total charge of $O=CO_2^{\delta-}$ is $-1.26e^-$ and charge on C is $+1.46$, leading to C 1s binding energy (BE) of $-269.45$ eV. The simulated BE value corresponds to 287.9 eV in the experimental observation (Fig. 2b). The configuration of the $O=CO_2^{\delta-}$ illustrated in top view is shown in Supplementary Fig. 3. The properties of the surface $O=CO_2^{\delta-}$ are summarized in Supplementary Note 3, and Supplementary Figs. 4 and 5. The simulated vibrational frequency data for $O=CO_2^{\delta-}$ are summarized in Supplementary Table 2.

We also investigated structures with vertical and horizontal $CO_3$ configurations on the Ag (111) surface (Supplementary Fig. 6). We find that the structure with one O bridging to the

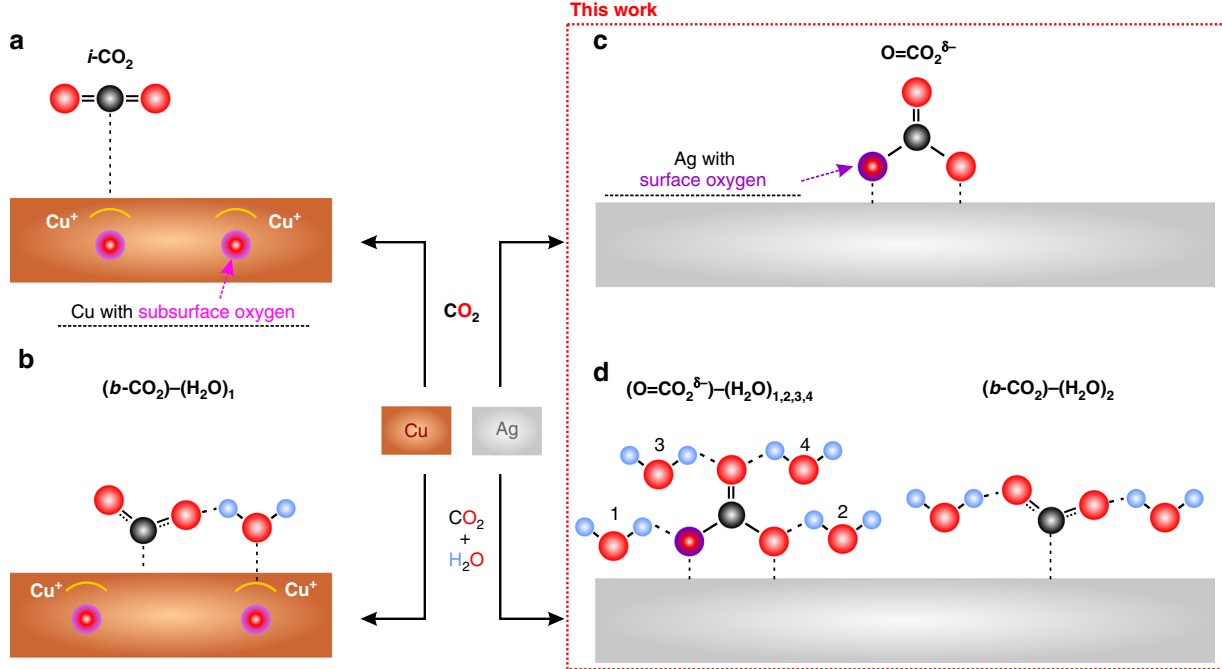

**Fig. 1** Overview of surface adsorptions and reactions of $CO_2$ on Cu and Ag surfaces under various conditions. **a, b** We earlier reported $CO_2$ adsorption on Cu (111) at 298 K both alone and in the presence of $H_2O$. These studies concluded that subsurface oxygen leads to a surface $Cu^+$ atom that stabilizes $l$-$CO_2$ sufficiently strongly to be stable at 298 K and 0.7 Torr (**a**). In the presence of subsurface O, we found that $H_2O$ adsorbs preferentially to the $Cu^+$ site while interacting sufficiently strongly with $CO_2$ to stabilize the $b$-$CO_2$, (through a hydrogen bond (**b**)) sufficiently to be stable at 298 K and 0.7 Torr total pressure. **c, d** Based on our new studies of adsorbed $CO_2$ on the Ag surface alone and in the presence of $H_2O$ at 298 K. We find that $l$-$CO_2$ is not stable on Ag surface even at $CO_2$ pressure of 0.3 Torr at 298 K. However, $CO_2$ reacts strongly with surface oxygen to form a carbonic acid like structure (**c**). This $O=CO_2^{\delta-}$ species can stabilize one to four adsorbed $H_2O$ through hydrogen bonding (**d**). Furthermore, $b$-$CO_2$ can also be stabilized by a pair of surface adsorbed $H_2O$ each forming a hydrogen bond with an O of $b$-$CO_2$ (**d**)

surface and two C–O bonds pointing up is not stable with $E_{ads} = +0.32$ eV. This starting structure rotates to form the stable bidentate species. We also examined the stability of the horizontal $CO_3$ configuration with three C–O bonds constrained to be parallel to the Ag surface. This configuration is not stable. The $CO_2$ bonding energy to form this horizontal structure is $\Delta E_{ads} = -0.34$ eV, $\Delta G = +0.13$ eV. Moreover the adsorption of $CO_2$ on the Ag (111) surface with an Ag vacancy induced by oxygen adsorption was examined and found to be unstable on this structure (Supplementary Fig. 7).

The adsorption states of $CO_2$ on various Ag surfaces at 298 K were monitored by C 1s APXPS. The pristine Ag surface shows no detectable carbon- and oxygen-based contamination (Supplementary Fig. 8), while dosing $O_2$ under different experimental conditions results in various oxygen coverages on Ag surface, that we monitor via the changes of the $O_{ad}$ peak intensity (the detailed characterizations of the surface are shown and discussed in Supplementary Note 4 and Supplementary Figs. 9 and 10).

We partition the C 1s spectra obtained on clean and oxygen-covered Ag surfaces into two parts. First, high binding energy region from 286–290 eV, showing the surface adsorbate, $O=CO_2^{\delta-}$ at 287.9 eV. $O=CO_2^{\delta-}$ is the only stable species on the Ag surface when exposed solely to $CO_2$ (no $H_2O$ is present), leading to a single C 1s peak in the adsorbate signal region of the APXPS spectra (Fig. 2b and Supplementary Fig. 11). Second, low binding energy region from 282 eV to 286 eV represents the surface reaction products from possible reactive carbon compounds (e.g., unsaturated hydrocarbons) from the chamber. The chemical species can be assigned as atomic C or carbide (283.0 eV), $sp^2$ C=C (284.2 eV), $sp^3$ C–C (285.2 eV), and C–O (H) (286.0 eV)[29–32] (Supplementary Fig. 12).

Formation of the carbonic acid-like species requires $O_{ad}$, which can be constituted from $O_2$ pre-dosing and $CO_2$ self-decomposition prior to the $CO_2$ adsorption. The experimental O 1s spectra shown in Supplementary Fig. 9 provide insight to elucidate the surface chemistry. The two peaks that represent two O atoms attached to Ag surface and the single O atom in the C=O bond, were used to fit the spectra. The energy difference between these two peaks was set as 0.7 eV based on the QM results. This leads to 2:1 peak intensity ratio. Thus the peak fitting of the experimental data supports the QM results. By further comparing the C and O signals, we obtain that the C:O atomic ratio are 1:2.85, 1:3.13, and 1:2.97 for adsorbates on pristine and low and high oxygen covered Ag surfaces, which are all close to 1:3, providing another strong evidence of the formation of $CO_3^{\delta-}$ structure on Ag surface.

Next, the adsorption of $CO_2$ on pristine Ag surface both alone and at the presence of 0.001 Torr $O_2$ at 298 K were investigated by recording the C 1s peak intensity as a function of gas dosing time (Fig. 3). The first spectrum was recorded after dosing $CO_2$ for 5 mins, which is the time period needed to reach 0.3 Torr pressure from the vacuum. In the case of the $CO_2$ adsorption, the adsorbate peak is negligible in the first spectrum recorded after 5 mins of $CO_2$ dosing, and it increases significantly as a function of increasing $CO_2$ dosing time, finally, it reaches equilibrate state after 60 mins gas adsorption. Adding $O_2$ with $CO_2$, even a ratio of 1:300, significantly promotes the process of $CO_2$ adsorption on metallic Ag. The adsorbate signal is strong in the first spectrum, and it does not change dramatically as a function of the increasing dosing time. During this dynamic process, the O:C atomic ratio were calculated to be around 3:1, validating the surface adsorbate of $CO_3^{\delta-}$ structure, as shown in Supplementary

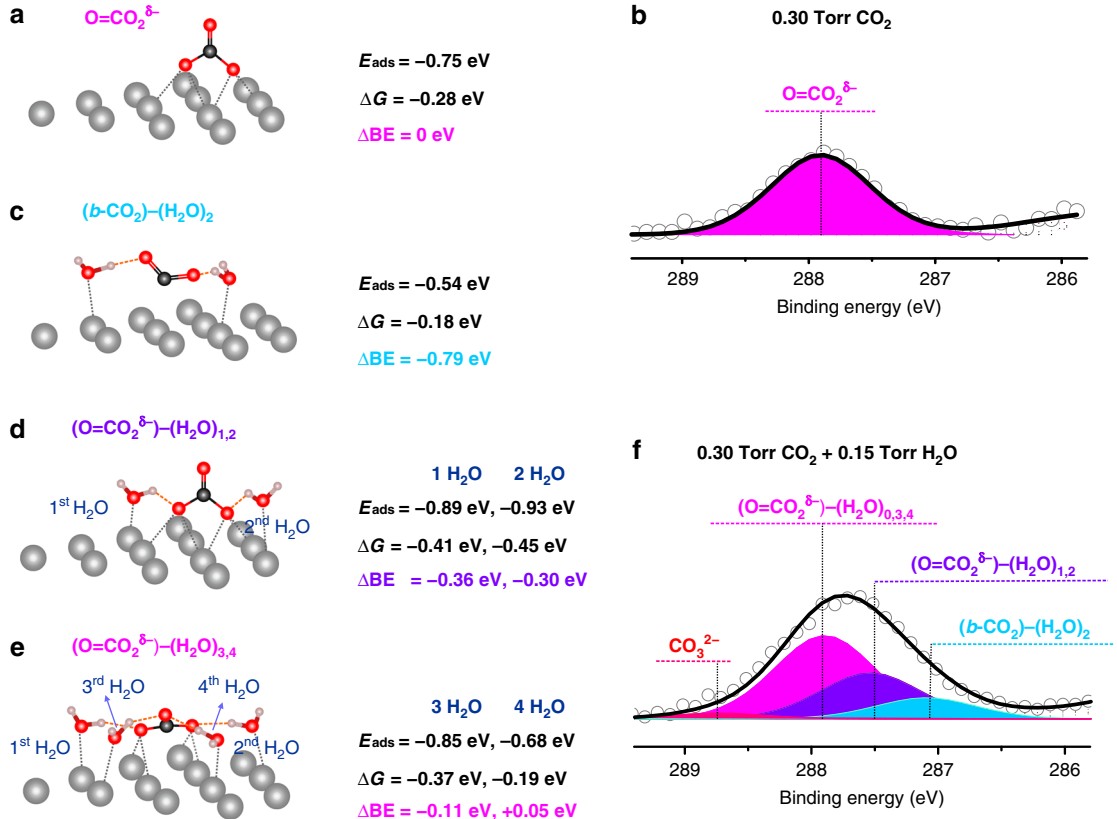

**Fig. 2** The QM predictions and experimental observations of Ag surface with $CO_2$ adsorption alone and in the presence of $H_2O$ at 298 K. **a** Predicted structures for $O=CO_2^{\delta-}$ on Ag surface. The $O=CO_2^{\delta-}$ C 1s peak BE has been set as the reference point for subsequent experiments with $H_2O$. **b** The C 1s APXPS spectra for Ag surfaces in the presence of 0.3 Torr $CO_2$ at 298 K. One single C 1s peak representing $O=CO_2^{\delta-}$ was observed. **c** $b$-$CO_2$ becomes stabilized by a pair of $H_2O_{ad}$ each forming a HB with an O of $b$-$CO_2$, leading to $\Delta G$ of −0.18 eV with respect to desorbing $H_2O$ and $CO_2$. **d, e** The adsorbed $O=CO_2^{\delta-}$ species stabilizes one or two $H_2O_{ad}$ via HBs to the $O_{ad}$ and two more water with HBs to the $O_{up}$. $O=CO_2^{\delta-}$ stabilizes the 1st, 2nd, 3rd, and 4th $H_2O$ on this site with $\Delta G$ of −0.41 eV, −0.45 eV, −0.37 eV, and −0.19 eV, respectively. **f** The C 1s APXPS spectra and the peak deconvolution results for Ag surfaces in the presence of 0.3 Torr $CO_2$ and 0.15 Torr $H_2O$ at 298 K. This deconvolution used the peak separations from the theory. The new surface adsorbates, $(O=CO_2^{\delta-})$-$(H_2O)_{1,2}$ and $(b$-$CO_2)$-$(H_2O)_2$, are observed experimentally in the APXPS measurements, showing up as the two new peaks at 0.4 eV and 0.8 eV, lower than the $O=CO_2^{\delta-}$ peak. The species $(O=CO_2^{\delta-})$-$(H_2O)_{3,4}$ do not lead to additional peaks, because they are located at position that overlaps with that of $O=CO_2^{\delta-}$

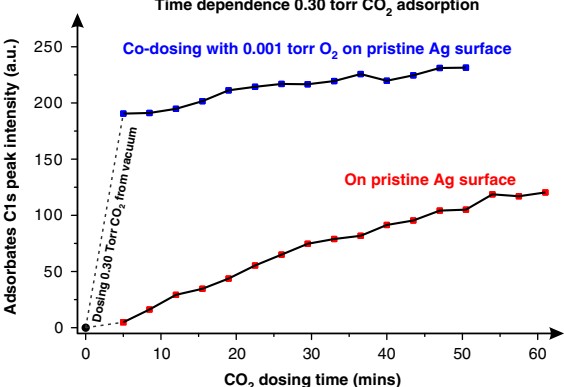

**Fig. 3** The C 1s signal of surface adsorbate increase as a function of $CO_2$ dosing time. The adsorbate signals for 0.3 Torr $CO_2$ adsorption alone and in the presence of 0.001 Torr $O_2$ were recorded at an increased dosing time, shown as red and blue points, respectively. A black line across the data point is used for the eye guidance. The peak intensity is the raw intensity without considering the signal decay due to introducing gases

Fig. 13. The largely accelerated process for the surface to reach equilibrium by adding $O_2$ is due to the formation of surface oxygen. Since $CO_2$ adsorption on clean (non-oxygen pretreated) Ag surface requires a $CO_2$ dissociation process prior to the formation of the final surface adsorbate, the dynamics of $O=CO_2^{\delta-}$ formation on clean Ag surface is slower than that with the oxygen co-dosed.

It is well known that during ambient pressure exposure of $CO_2$, possible residual reactive carbon compounds (e.g., unsaturated hydrocarbons) can be desorbed from the chamber. Thus, due to the slow surface reaction of $CO_2$ on the clean Ag surface could lead to a larger possibility for the Ag surface to be exposed to unsaturated hydrocarbons that can lead to the formation of the $sp^2$ carbon species. After the surface acquires surface $O_{ad}$ (Supplementary Fig. 9) or is co-dosed with $O_2$ (Fig. 3), $CO_2$ can react directly on the surface to form $O=CO_2^{\delta-}$. This suppresses surface carbon formation as evident in the decrease of the surface carbon (mainly the $sp^2$ $C=C^{29-32}$) C 1s signals (Supplementary Fig. 12), resulting in more available surfaces sites to increase the amount of adsorbed $O=CO_2^{\delta-}$ (Fig. 3 and Supplementary Fig. 11).

In addition, we made an estimate of the surface coverage by calculating the Ag and O atomic ratio, and the $O=CO_2^{\delta-}$:$Ag_{suf}$

ratios, which we found to be around 0.4:1, 0.6:1, and 0.7:1. This indicates that the reaction between surface O and Ag to form $O=CO_2^{\delta-}$ happens at surface majority sites, justifying the use of the Ag (111) model in the this study.

**$CO_2$ adsorption on Ag surfaces in the presence of $H_2O$**. The QM studies find that the $l$-$CO_2$ configuration on Ag surface is not stable even considering the possible promotion effects of both $O_{ad}$ and adsorbed water ($H_2O_{ad}$) (Supplementary Fig. 2). Adding $H_2O$ to the surface with $O=CO_2^{\delta-}$ formed from $g$-$CO_2$ leads to two groups of structures stable at 298 K and the applied pressures (Fig. 1d). First, a pair of surface $H_2O$ stabilizes $b$-$CO_2$ on the Ag surface by forming two HBs between the $H_2O_{ad}$ and $CO_2$ (Fig. 2c). Second, $O=CO_2^{\delta-}$ can stabilize up to 4 $H_2O$ molecules through formation of HBs to the surface bonds of $O=CO_2^{\delta-}$. The 1st and 2nd $H_2O_{ad}$ each forms a HB to one $O_{ad}$ bonded to the surface (Fig. 2d), while adding the 3rd and 4th $H_2O$ force the $C=O_{up}$ bond to rotate from being perpendicular to the surface to being nearly parallel to the surface, allowing the formation of HB from a 3rd and 4th surface $H_2O_{ad}$ to the two $sp^2$ lone pairs on the $C=O_{up}$ unit (Fig. 2e and Supplementary Figs. 3 and 4). From QM predictions, the 1st and 2nd $H_2O_{ad}$ shift the C 1s from $-269.45$ eV to $-269.09$ eV and $-269.15$ eV, while the 3rd and 4th $H_2O_{ad}$ shift the C 1s back to $-269.34$ eV and $-269.50$ eV, nearly the same BE's as for no $H_2O_{ad}$ bonding (Fig. 2f and Supplementary Fig. 5). Considering that the $O=CO_2^{\delta-}$ and surface water stabilize each other through HB, an increase in the surface adsorbate coverage when dosing $CO_2$ in the presence of $H_2O$ is expected. This was experimentally observed as a dramatic adsorbate signal increase of C 1s spectra compared to that from the adsorption of $CO_2$ alone (Supplementary Figs. 11 and 12). Moreover, the simulated vibrational frequency data for $(O=CO_2^{\delta-})-(H_2O)_{1-4}$ and $b$-$CO_2-(H_2O)_2$ are summarized in Supplementary Table 2.

**The tunability of $CO_2$ adsorption on metal surfaces**. Activating inert $CO_2$ to $b$-$CO_2$ requires both a change of the geometric molecular structure and accommodation of extra charge. For Cu, $b$-$CO_2$ is stabilized by a subsurface O combined with a single surface adsorbed $H_2O_{ad}$ while for Ag it is stabilized by two adsorbed $H_2O_{ad}$. The $b$-$CO_2$ with surface $H_2O$ configuration leads to a similar amount of charge transferred directly from the metal catalyst to the C for both Cu and Ag. Interestingly, the $b$-$CO_2$ on Ag and Cu surfaces show similar charge distribution (calculated by performing Bader Charge Analysis on optimized structures[33–35]) but different C 1s binding energy peak positions. This may be ascribed to the increased final state screening effect of Cu on surface $b$-$CO_2$ due to the smaller distance between the surface adsorbate and the metal substrate (2.55 Å for C–Ag vs. 1.69 Å for C-Cu)[36]. The direct Ag–C interaction in $(b$-$CO_2)-(H_2O)_2$ leads to a $-0.67e^-$ charge accumulating on the adsorbed $CO_2$ molecule which is larger than the $-0.3e^-$ for the $O=CO_2^{\delta-}$ configuration (compared to $O_{ad}$) (Fig. 4). Moreover, adding surface $H_2O$ leads to additional charge redistribution through the hydrogen bonding (Fig. 4). Attaching more water to $O=CO_2^{\delta-}$ decreases the total charge on adsorbates, while the 1st $H_2O$ decreases the charge on C atoms to $+1.27$ and the 2nd to 4th shift it back to $+1.48$, nearly the same as for no $H_2O$ (Fig. 4). The charge distribution on the various surface adsorbates are detailed in Supplementary Note 3.

This work highlights that the charge transfer configurations are responsible for the tunability of $CO_2$ adsorption on the metal catalyst surface. These results suggest two modes for stabilizing adsorbed $CO_2$. In the case of Cu, a subsurface O provided a positive $Cu^+$ on the surface that stabilized water molecule

sufficiently to stabilize $b$-$CO_2$. This mechanism has been studied previously[37].

For Ag there is no subsurface O, but the surface $O_{ad}$ promotes the formation of surface carbonic acid-like species, $O=CO_2^{\delta-}$, which leads to a very different reaction mechanism for Ag than for Cu. This new insight requires re-examining the subsequent steps of reactions to activate $O=CO_2^{\delta-}$ to form products and how this depends on surface structure, solvent, pH, applied potential, the presence of anions and cations, and alloying with nonmetals (S, P, N, Cl) that might change the local charges and structures.

**Proposed $CO_2$ reduction reaction pathway on Ag and Cu**. The $CO_2$ adsorption on Ag contrasts dramatically from the results on Cu (Supplementary Table 3) providing possible explanations for why these metal catalysts have very different $CO_2$ reduction performances. For Cu our full explicit solvent QM calculations for the initial step of $CO_2$ to CO found that hydrogen bonding with the explicit solvent forms a similar $b$-$CO_2$ stabilized by two surface $H_2O$[37]. In that case, the next step is for one of these two $H_2O$ molecules to transfer an H to form the HOCO intermediate plus $OH_{ad}$ and then a second surface $H_2O$ transfers an H to the OH of HOCO to form $H_2O$ plus $OH_{ad}$, leading to $CO_{ad}$, (this general reaction pathway is depicted in Fig. 5a).

For Ag with $(b$-$CO_2)-(H_2O)_n$, Fig. 5a shows that the QM predicted free energy barrier is 0.99 eV on Ag for protonating the complex of $b$-$CO_2$ with two $H_2O$ to form HOCO* plus OH* and $H_2O$ (Supplementary Fig. 14), leading to a total barrier of hydrogenation of $CO_2$ to HOCO* of $(-0.18)+(0.99)=0.81$ eV (Fig. 5a). This energy barrier is comparable to that on Cu, which is 0.80 eV[37].

Surprisingly for Ag with $(O=CO_2^{\delta-})-(H_2O)_n$ clusters we find a different mechanism that is significantly more favorable. The discovery that $(O=CO_2^{\delta-})-(H_2O)_n$ is a stable surface cluster is most unprecedented, differing dramatically from our previous understanding of $CO_2$ on a metal surface, which essentially all involve $(b$-$CO_2)-(H_2O)_n$ configurations[38–42].

We used QM to discover the mechanism of activation for the carbonic acid-like species on Ag. We find that the first step is for the $H_2O$ hydrogen bonded to the surface O to transfer an H to form the $(C=O)(O)(OH)$ intermediate plus $OH_{ad}$, as shown in Fig. 5b. The QM energy barrier is 0.62 eV, which is dramatically lower than the value of 0.80 eV for Cu, perhaps explaining the faster rate for Ag. Thus the barrier of hydrogenation of $CO_2$ to OCOOH* of $(-0.28)+(-0.41)+(-0.45)+(0.62)=-0.52$ eV (Fig. 5b and Supplementary Fig. 14). This energy barrier is much smaller than for Cu. In particular, it is important to note that the energy levels of all the reaction steps starting with $O=CO_2^{\delta-}$ are negative. This suggests that we might be able to see this reaction in APXPS by simply increasing the temperature. These results predict that the most energetically favorable reduction reaction pathway to hydrogenate $CO_2$ to HOCO* involves the $O=CO_2^{\delta-}$ configuration present only on Ag surface. This process is unprecedented and has never even been previously speculated. This result raises numerous questions about subsequent steps that will drive many new experimental and theoretical studies to determine the implications. Future studies will include the operando spectroscopic characterizations of these adsorbates under external potentials, and we will predict the Tafel slope to compare with previous experimental observations and to gain more insights into the new mechanism.

**Discussion**

Our studies have established a comprehensive but totally new picture of the first steps of $CO_2$ activation on Ag. The dramatic

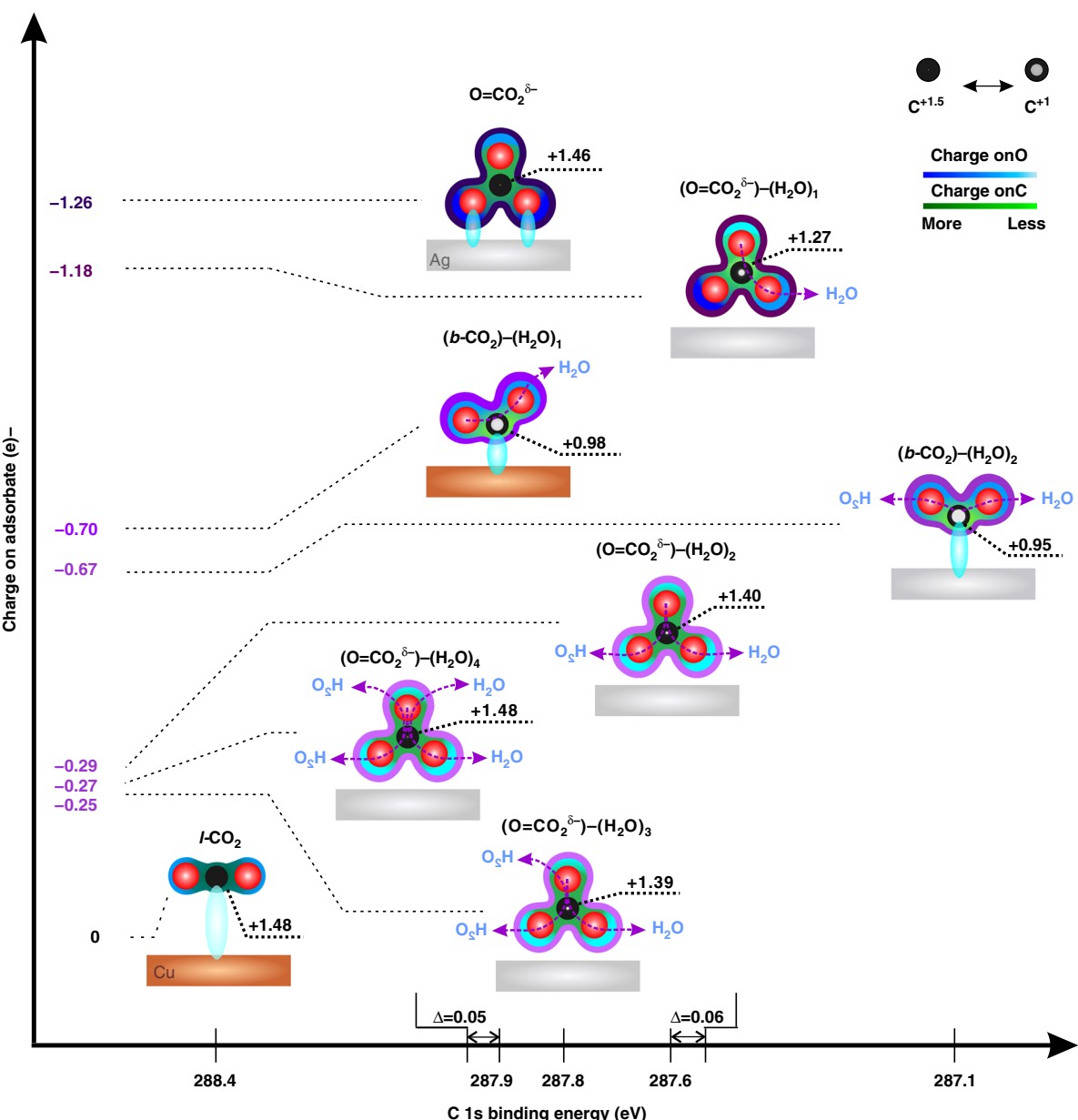

**Fig. 4** The electronic properties of various surface adsorbates on Ag and Cu. The charge distribution on the C, O and the adsorbates are summarized, with the corresponding C 1s BE revisited. The various configurations of the adsorbates on the surface modify the charge transfer process, leading to different charge distribution on the adsorbates. Compared to $l$-$CO_2$ (only observed on Cu surface), $CO_2$ in the bent configuration exhibits extra charge accumulation. $b$-$CO_2$ is stabilized on Ag only with two surface $H_2O$ but the charge distribution is similar to $b$-$CO_2$ on Cu surface. However, their different distances to the Ag and Cu surface lead to different C 1s peak BE's. With the formation of the first two HBs to surface $H_2O$, the total charge on $O = CO_2^{\delta-}$ decreases, which decreases the C 1s BE by ~0.30 eV. But adding the 3rd and 4th $H_2O$ with HB to the $C = O_{up}$ of the $O = CO_2^{\delta-}$ increases the charge, shifting the BE back to 0.05 eV above the peak for no $H_2O$. Thus the predicting C 1s shifts and charge distribution on surface adsorbates are fully consistent with the experimental observed C 1s BEs. These observed differences show the tunability of $CO_2$ adsorption on the metal surfaces

differences with Cu show how interactions between adsorbate and catalyst can be altered by tuning the charge transfer between them through changing the adsorption sites, configuration, and by introducing surface co-dosing adsorbates. These findings provide fresh insights about $CO_2$ adsorption species and the initial steps of $CO_2$ reduction mechanism on Ag surfaces. It is dramatically different from those on Cu surfaces, where $l$-$CO_2$ leads to $b$-$CO_2$ and then directly to $CO_2$ reduction[32].

Using synergistic experimental and theoretical analyses, we show that Cu and Ag operate entirely differently for the first step of activating $CO_2$, even though the product CO is the same. We

find that surface O (from $O_2$ pre-dosing and $CO_2$ self-decomposition) interacts with $g$-$CO_2$ to form a carbonic acid like intermediate $O = CO_2^{\delta-}$, the only stable species on Ag surface (exposed to $CO_2$ only). Adding $H_2O$ and $CO_2$ then leads to attaching up to four water on $O = CO_2^{\delta-}$. In addition, two water stabilize $b$-$CO_2$ on the Ag surface, which for Cu is the intermediate on the way to forming CO. On Ag we find a very different and much more favorable mechanism involving the $O = CO_2^{\delta-}$, one that has not been suggested or observed previously. This raises numerous questions about the subsequent steps that could motivate the exploration of new chemistries.

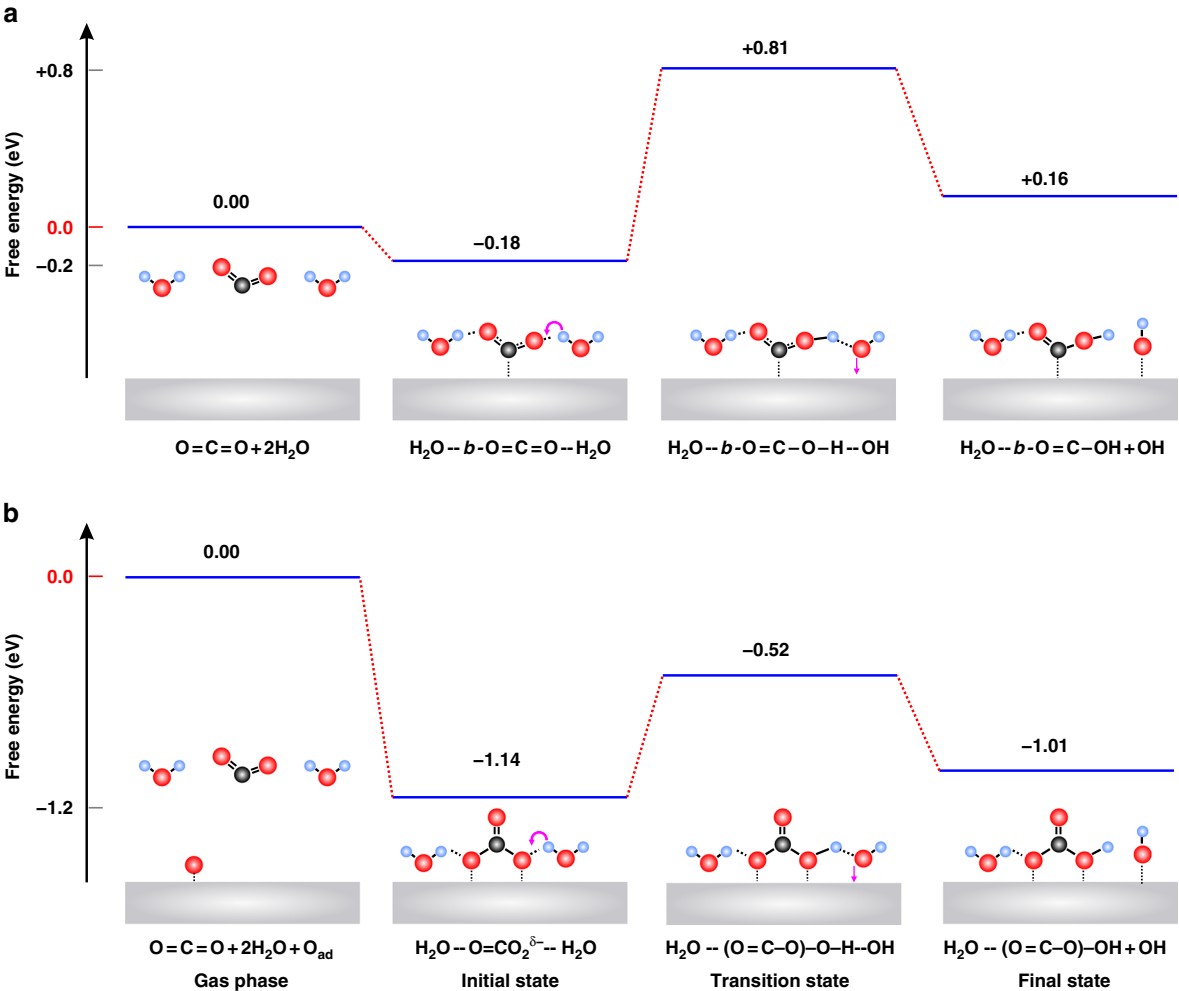

**Fig. 5** The QM predicted kinetic pathway for the $CO_2$ hydrogenation process from full explicit solvent calculations. **a** The reaction pathway starting with $b$-$CO_2$, the energy level of each step is referenced to $g$-$CO_2$ and $g$-$H_2O$; **b** The reaction pathway starting with $O=CO_2^{\delta-}$, the energy level of each step is referenced to $g$-$CO_2$, $g$-$H_2O$ and surface $O_{ad}$. The first step was shown in Fig. 2, representing the stable adsorption configuration observed on the catalyst surface. The energy barrier information obtained from our climbing image nudged elastic band (NEB) calculations are detailed in Supplementary Fig. 14 We consider here the case of $O=CO_2^{\delta-}$ with $2H_2O$ to compare directly with $b$-$CO_2 + 2H_2O$

These studies emphasize the power from combining BE, vibrational frequency, APXPS with QM for discovering the fundamentals underlying $CO_2$ reduction. These unexpected findings will stimulate new thinking about the $CO_2$ reduction reactions on metal surfaces, suggesting that stabilization of various surface adsorption configurations can be controlled through additives or alloying along with externally applied potentials to control the reaction processes.

## Methods

**QM calculations**. All calculations were carried out with the Vienna Ab initio Simulation Package (VASP)[43]. We established that an energy cutoff of 500 eV leads to converged forces. The K-point sampling was chosen to be $3 \times 3 \times 1$. All calculations include spin-polarization. We used the Perdew-Burke-Ernzerhof (PBE) flavor of Density Functional theory (DFT), including the D3 (Becke Johnson)[44] empirical corrections for long range London dispersion (van der Waals attraction)[45].

The PBE-D3(BJ) level of DFT leads to a calculated lattice parameter of $a =$ 4.012 Å for the bulk Ag structure at 0 K, slightly smaller than the experimental value 4.085 Å at 298 K[46]. We used experimental lattice parameter 4.085 Å to construct a two-dimensional periodic slab with four layers of Ag atoms each of which consists of a $(4 \times 4)$ unit cell (16 surface Ag per cell). We include 15 Å of vacuum in the $z$ direction to minimize possible interactions between the replicated cells. The top two layers are relaxed while the bottom layers are kept fixed.

This level of QM has been validated recently for several systems. Thus references carried out systematic studies for the oxygen reduction reaction (ORR, $O_2 + protons \rightarrow H_2O$) on Pt (111) using the same PBE-D3 level as in this paper[47].

Including 5 layers of explicit solvent in QM metadynamics on all reaction steps, comparisons could be made to experimental activation barriers for two values of the external potential. In both cases the calculated activation barriers were within 0.05 eV of the experiment[48–51].

Previous calculations for the $CO_2$ reduction reaction on Cu (100) using the same level of theory obtain an activation energy within 0.05 eV of experiment. This same level of theory has also led to similar accuracy for the oxygen evolution reaction on $IrO_2$ and for onset potentials on Cu (111)[52,53].

Calculations for the gas phase molecules used the PBE functional (as implemented in Jaguar) with the D3 empirical correction for London dispersion[54]. To obtain the total free energy, $G = H - TS$, for the gas molecules at temperature T, we add to the DFT electronic energy (E), the zero-point energy (ZPE) from the vibrational levels (described as simple harmonic oscillators), and the specific heat corrections in the enthalpy from 0 to T. The entropy (S), as a sum of vibrational, rotational and translational contributions, are evaluated from the same levels. To correct the free energy for pressure, we assume an ideal gas and add $RT \times \ln(P_2/P_1)$ with a reference pressure of $P = 1$ atm. For example, $CO_2$ gas at room temperature and 1 atm would have a free energy correction of $-0.25$ eV, including ZPE (0.32 eV), translational entropy contribution ($-0.42$ eV), rotational entropy contribution ($-0.15$ eV) and almost negligible vibrational entropy contribution ($-0.003$ eV). All calculations assume the current experimental condition: $P(CO_2) = 0.3$ Torr, and $P(H_2O) = 0.15$ Torr.

After the gas molecules adsorbed on the metal surface, their rotational and translational degrees of freedom are reduced to vibrational modes. The vibrational frequencies for surface adsorbents are calculated by allowing the adsorbed molecules and the top layer of metal to relax, with the bottom layers fixed. For these phonon calculations we used $10^{-6}$ eV energy convergence threshold to obtain reliable phonon frequencies (no negative eigenvalues.) To obtain the Free energy,

$G = H - TS$, for the various equilibrium configurations, we used density functional perturbation theory (DFPT) to calculate the phonon density of states, which was used to calculate the ZPE, the temperature correction to the enthalpy, and the vibrational contributions to the entropy.

There are two ways of calculating the change in core level energies implemented in VASP[43]. The simpler option (ICORELEVEL = 1) calculates the core levels in the initial state approximation, which involves recalculating the KS eigenvalues of the core states after a self-consistent calculation of the valence charge density. The second option (ICORELEVEL = 2) is more involved. In this case, electrons are removed from the core and placed into the valence. Our previous studies found that the ICORELEVEL = 1 leads to relative binding energy shift in good agreement with experimental XPS[17].

**In-situ ambient pressure X-ray photoelectron spectroscopy measurements.**
Ambient pressure XPS measurements were performed at Beamline 9.3.2 of the Advanced Light Source, Lawrence Berkeley National Laboratory[55]. The beamline has station consisted of a load lock chamber with base pressure of $\sim 5 \times 10^{-8}$ Torr for sample loading; a preparation chamber with base pressure of $\sim 1 \times 10^{-9}$ Torr for sample preparation, and a main chamber for sample characterization under ambient pressure condition. The beamline provides beams with a photon energy range of 200–800 eV.

The pristine Ag surface was in-situ prepared in the vacuum chamber by repeated argon sputtering (2 keV, 60 min) and vacuum annealing (900 K, 60 min), leading to a clean surface with no detectable carbon- and oxygen- based contamination. The oxygen covered Ag surfaces were prepared by annealing the samples at 430 K at 0.04 Torr $O_2$ for 5 min, and 0.06 Torr $O_2$ for 15 min, respectively.

During the APXPS measurements performed at 298 K, $CO_2$ partial pressure was kept at 0.3 Torr for $CO_2$ adsorption, whereas the total pressure was kept at 0.45 Torr with 0.3 Torr $CO_2$ and 0.15 Torr $H_2O$. The purities of the dosing gases ($CO_2$, $H_2O$) were in-situ monitored by a conventional quadrupole mass spectrometer to ensure no additional gas cross-contamination (especially, the CO and $H_2$ gases).

The XPS spectra were collected at an incident photon energy of 670 eV, in the following order: a low-resolution survey with a binding energy of 600 eV to –10 eV, then high-resolution scans of O 1 s, C 1 s and valence band. The inelastic mean free path (IMFP) for the photoelectrons was below 0.9 nm for all the spectra collected. For each condition, samples were equilibrated for at least 30 mins before the measurement. By taking spectra at different sample spots and comparing spectra before and after beam illumination for 2 h, we found beam damage on the sample is negligible during the measurements.

## Data availability
The data that support the findings of this study are available from the corresponding authors upon request.

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

## Acknowledgements

This work was supported through the Office of Science, Office of Basic Energy Science (BES), of the US Department of Energy (DOE) under Award DE-SC0004993 to the Joint Center for Artificial Photosynthesis, DOE Energy Innovation Hubs. The Advanced Light Source is supported by the Director, Office of Science, Office of BES, of the US DOE under Contract DE-AC02-05CH11231. H.Y. and H.S. gratefully acknowledge China Scholarship Council (CSC, No. 201608320161 and No. 201706340112) for financial support. This work used the Extreme Science and Engineering Discovery Environment (XSEDE), which is supported by National Science Foundation grant number ACI-1548562. Y.Y. and E.J.C. were partially supported by an Early Career Award in the Condensed Phase and Interfacial Molecular Science Program, in the Chemical Sciences Geosciences and Biosciences Division of the Office of Basic Energy Sciences of the U.S. Department of Energy under Contract No. DE-AC02-05CH11231.

## Author contributions

Y.Y., J.Y., W.A.G. III and E.J.C. designed the experiments. Y.Y., H.S., K.J.L. and E.J.C. performed the APXPS experiments. H.Y., J.Q., T.C. and H.X. conducted the theoretical computations. Y.Y., H.Y., J.Q., J.Y., W.A.G. III and E.J.C. analyzed the data and wrote the manuscript. All authors contributed to the overall scientific interpretation and edited the manuscript.

## Additional information

**Competing interests:** The authors declare no competing interests.

