## [Peer Review File · Nature Communications]

Reviewers' comments:

Reviewer #1 (Remarks to the Author):

This manuscript describes quantum mechanical calculations and ambient pressure XPS experiments regarding CO₂ adsorption onto a Ag surface. The authors claim the formation of a O=CO₂^{δ-} species which has previously not been reported, and surface oxygen on silver promotes the formation of this species. These observations are contrasted to copper, and a new first step mechanism for CO₂ reduction on silver is proposed to explain the CO₂ reduction reactivity differences. This work is novel and provide unique insights, which could be considered for a publication in Nature Communications after addressing the following comments.

Technical comments:

1. The authors should show the O1s XPS spectra for pristine Ag after dosing CO₂ in Supplementary Figure 5. For a clean Ag surface, if the CO₂ dissociates to form CO and the O=CO₂^{δ-}, then the O1s spectra should show O=CO₂^{δ-} with no Oad.
2. The authors should provide more detail on how the charges in Figure 3 are calculated.
3. Surface-bound Oads is proposed to facilitate the adsorption of CO₂. However, would a significant amount of Oads be present under the reductive potentials during CO₂ electrolysis?
4. The authors proposed a new CO₂ reduction reaction mechanism on Ag and supported it with computational modeling. What is the predicted Tafel slope based on the new mechanism? Is it consistent with previous experimental observations? Some discussions will be highly appreciated.

Reviewer #2 (Remarks to the Author):

The manuscript describes computational and experimental work on CO₂ adsorption and reduction on Ag(111). The results are compared to previous results by the Authors on Cu(111). The experiments report AP-XPS of the C 1s binding energy while the computations have been performed within the density functional theory using a state-of-the-art code. The main conclusions are that H₂O strongly affects the activation and that the mechanisms on Cu and Ag are different.

There is currently considerable interest in the activation of CO₂ and the idea to compare different metal surfaces is interesting and important. This kind of work has the potential to enhance the understanding and ultimately aid the development of catalysts with favorable performance. The manuscript is interesting and should be considered for publication after the Authors have addressed the following three major points:

1. A new mechanism is suggested for CO₂ adsorption on Ag(111). In particular, CO₂ is proposed to adsorb on oxygen covered Ag(111) by the formation of a charged O-CO₂ species. There are issues with this suggestion. Adsorbed oxygen is at the applied conditions unstable with respect to the formation of "surface oxide" structures. There are a range of different structures that may form depending on growth conditions. The p(4x4) is perhaps the mostly discussed structure in the literature. The Authors should consider CO₂ adsorption on such structures.
2. A second issue is that it has previously been suggested that CO adsorption on p(4x4) leads to facile carbonate formation, see Phys. Rev. B, 2011, 84, 115430. The C 1s binding energy reported for the carbonate in the PRB paper is the same as reported for the O-CO₂ in the present manuscript. It is unclear why the Authors rule out the formation of a carbonate. The Authors need to strengthen the arguments with respect to the existence of an O-CO₂ species. The presented experimental result (only

C 1s binding energy) is not convincing. XPS spectra of Ag and O should make it clearer which species that is present and the state of the surface. Another possibility would be to perform IR-measurements.

3. The results on Ag(111) are put in perspective by comparison with CO₂ adsorption on Cu(111). In the case of Cu(111), it is argued that CO₂ adsorption is stabilized by subsurface oxygen. However, this is a controversial and recent calculations indicate that subsurface oxygen is highly unstable and would convert to surface oxygen during typical reaction conditions. See Garza et al. J. Phys. Chem. Lett., 2018, 9, 601. The Authors need to discuss this point as subsurface oxygen appears to be unlikely.

Reviewer #3 (Remarks to the Author):

The authors report results of an experimental and computational study in which they claim that CO₂ adsorption on Ag produces a carbonic acid species of the form, O=CO₂d-, by coordinating with a chemisorbed O-atom, and that this species also forms in the presence of H₂O and hydrogen-bonds with co-adsorbed H₂O molecules. The authors present DFT results which predict that the carbonic acid species is stable and energetically-preferred over other configurations. They also present AP-XPS (ambient pressure X-ray photoelectron spectroscopy) results which they claim support their assignments. I find that their AP-XPS spectra are not convincing in supporting the proposed assignments, and identify several issues that raise significant doubts about the interpretation of their experimental results. The paper is also quite difficult to follow in my opinion. I note that the Supplementary Information is nearly twice as long as the main manuscript, and the authors repeatedly direct the reader to the SI for important information. At no point in the main manuscript or the SI do the authors clearly show how the experimental AP-XPS spectra agree unambiguously with the DFT predictions. Further, the reported C 1s spectra (SI, Figure 6) indicate that their polycrystalline Ag foil was contaminated with several, unknown carbon species. As such, assigning one of the several C 1s peaks to a O=CO₂d- species is highly questionable. This paper is unpublishable in its current form. More specific comments are as follows:

1) Experiments were performed using a polycrystalline Ag foil whereas the DFT calculations modelled a Ag(111) surface. The foil exposes many types of surface sites that are not reproduced by the Ag(111) model. As such, comparison between the experimental and computational results is questionable at the outset.

2) The authors state that DFT predicts that the formation of subsurface oxygen in Ag(111) is energetically unfavorable, and they thus omit subsurface oxygen from further consideration in their study, including when they assign O 1s peaks in the experimental data. I note, however, that the formation of subsurface oxygen in Ag and its role in mediating surface chemistry on Ag has been reported by numerous authors for many years. The authors make no mention of the vast literature on this topic.

3) The absolute C 1s binding energies computed using DFT are on the order of 20 eV lower than experimental values. The authors claim that differences in the C 1s core levels that are less than 0.5 eV are accurate. I find this claim highly questionable, even though they cite a paper in the SI that they state supports the idea.

4) The authors predict with DFT that the O=CO₂d- species features two distinct O-atoms, in a 2:1 ratio, that have different O 1s binding energies. However, they do not present experimental evidence to support this prediction.

5) In the SI, the authors report O 1s spectra of clean and oxygen-exposed Ag foil and deconvolute the

spectra into three peaks (528.5, 530.3, 531.5 eV). They assign these peaks to chemisorbed atomic oxygen, the O=CO₂d⁻ species and OH species. How can the peak at 530.3 eV originate from the O=CO₂d⁻ species when no such species should be present? This is illogical because the experiments were performed on O-exposed Ag, without the presence of CO₂, and actually suggest that the peak at 530.3 eV arises from a surface carbon species other than the O=CO₂d⁻ species. Also, are they claiming that OH is a contaminant? Again, the O 1s spectra are reported for O-exposed Ag foil so OH could only be caused by contamination, which is clearly a possibility (see below). I note, however, that subsurface O in Ag has been reported to give a distinct O 1s peak in prior studies.

6) The authors assign a C 1s peak at 289.7 eV to the O=CO₂d⁻ species on Ag foil. However, they do not present clear evidence to support this assignment. It is insufficient and unconvincing to assert that their assignment is correct only because DFT predicts that this species forms. As far as I can tell, the only spectral evidence that they claim supports their peak assignment is that they can deconvolute the C 1s peak from (presumably) the O=CO₂d⁻ species into three components when H₂O is present with CO₂. This is unconvincing for several reasons. A key reason is that they observe several other C 1s peaks as well, indicating that their Ag sample is highly contaminated with unknown carbon-containing species. Thus, the peak at 289.7 eV could arise from numerous species, not considered by DFT. An inability to correlate the C 1s feature with the O 1s feature raises further doubts about the assignment, though the carbon contamination and polycrystalline surface are much larger concerns.

7) The authors mention the CO₂ and H₂O partial pressures used in their experiments but do not mention the sample temperature during these exposures. They also mention that they vacuum annealed the Ag foil but do not mention the annealing temperature. Including such information is imperative and its omission is very surprising.

8) In the SI, the authors report intensity ratios of the O 1s to Ag 3d peaks as a way to differentiate surfaces with differing amounts of oxygen. At the least the authors should apply sensitivity factors to account for differences in the O 1s and Ag 3d photoelectron cross sections. However, a more appropriate way to quantify coverage is to collect spectra from a surface with a known coverage of adsorbate. The current data does not provide a reliable estimate of the adsorbate amounts.

9) Supplementary Figure 6 shows C 1s spectra obtained from the Ag foil in the presence of CO₂. The C 1s spectra exhibit multiple peaks that are consistent with surface contamination by adventitious carbon as well as various C-O moieties. The corresponding intensities of these peaks are significant and in several cases larger than the peak that they claim arises from the O=CO₂d⁻ species. Thus, the authors are basing their interpretations about the formation of an (alleged) specific species on spectra obtained from a highly carbon-contaminated, polycrystalline Ag foil. Such an interpretation is highly questionable.

NCOMMS-18-26430-T.

"Dramatic differences in CO₂ adsorption and initial steps of reduction between Ag and Cu"

Reviewer #1 (R1):

This manuscript describes quantum mechanical calculations and ambient pressure XPS experiments regarding CO₂ adsorption onto a Ag surface. The authors claim the formation of a O=CO₂^{δ-} species which has previously not been reported, and surface oxygen on silver promotes the formation of this species. These observations are contrasted to copper, and a new first step mechanism for CO₂ reduction on silver is proposed to explain the CO₂ reduction reactivity differences. This work is novel and provide unique insights, which could be considered for a publication in Nature Communications after addressing the following comments.

Response to Reviewer #1. Thank you for your helpful comments and suggestions. We in particular appreciate your strong support for publication in Nature Communications. Our response is in blue bold face below.

Technical comments:

The authors should show the O1s XPS spectra for pristine Ag after dosing CO₂ in Supplementary Figure 5. For a clean Ag surface, if the CO₂ dissociates to form CO and the O=CO₂^{δ-}, **then the O1s spectra should show O=CO₂^{δ-} with no O_{ads}.**

Author Reply: Thank you for this suggestion, we have added the O1s spectra (Figure R1) to Supplementary Fig. 6 (previously denoted as Supplementary Figure 5) to provide the direct comparison of O1s spectra before and after CO₂ adsorption. Figure R1 (a) shows the pristine and oxygen pre-treated Ag surface prior to the CO₂ dose. Two main features appear during the oxygen pre-dosing. A small peak representing adsorbed oxygen (O_{ads}) on Ag appears at 528.5 eV. In addition, we observe a peak at 530.3 eV corresponding to the surface O=CO₂^{δ-} that forms on the surface from reaction with residential CO and CO₂ gases in the chamber.

After dosing CO₂ (Figure R1 (b)), we found that the O_{ads} feature vanished for all the samples due to the surface reaction happening between O_{ads} and CO₂. We hope this answered the reviewer's question. We have included this information in the Supplementary materials.

Figure R1. O1s spectra of pristine and oxygen-covered Ag surfaces before and after CO₂ adsorption. This is now a part of Supplementary Fig. 6 (previously denoted as Supplementary Figure 5)

2. The authors should provide more detail on how the charges in Figure 3 are calculated.

Author Reply: The charges for Figure 4 (previously denoted as Figure 3) are calculated by performing Bader Charge Analysis on optimized structures, which is the standard method for plane wave calculations with VASP

We have amended the figure caption to now include the following underlined sentence: “The charge distribution (calculated by performing Bader Charge Analysis on optimized structures) on the C, O and adsorbates...” and included the following references:

1)W. Tang, E. Sanville, and G. Henkelman, A grid-based Bader analysis algorithm without lattice bias, J. Phys.: Compute Mater. 21 084204 (2009).

2)E. Sanville, S. D. Kenny, R. Smith, and G. Henkelman, An improved grid-based algorithm for Bader charge allocation, J. Comp. Chem. 28 899-908 (2007).

3)G. Henkelman, A. Arnaldsson, and H. Jónsson, A fast and robust algorithm for Bader decomposition of charge density, Comput. Mater. Sci. 36 254-360 (2006).

3. Surface-bound Oads is proposed to facilitate the adsorption of CO₂. However, would a significant amount of Oads be present under the **reductive potentials during CO₂ electrolysis?**

Author Reply: In this work, we focused on the adsorption and reaction of CO₂ on the Ag surface, and we investigated the first step of hydrogenation. We show that O_{ads} facilitates the adsorption of CO₂ to form O=CO₂^{δ-} on Ag, which leads to a new favorable hydrogenation process. For all these processes, no external potential is applied. Therefore, we currently do not have a clear answer to this question. Future experiments using operando studies with external potential applied we do believe will help to answer this question.

Our observations from the literature based on additional studies performed using Cu electrodes are as follows.

It has been reported that surface oxygen and subsurface O on the Cu electrode is not completely removed at reducing potentials. Some studies indicate that keeping the oxide-derived copper electrodes at -1.15 V vs RHE for 1h did not change the oxygen content and distribution at the topmost 1-2 nm amorphous layer. These results were performed at *in-situ* or quasi *in-situ* conditions, providing strong evidence that subsurface/surface oxygen is stable for an extended time under reductive change. (*Ref: Nature and distribution of stable subsurface oxygen in copper electrodes during electrochemical CO₂ Reduction, J. Phys. Chem. C, 2017, 121 (45), 25003–25009; Subsurface Oxygen in oxide-derived copper electrocatalysts for carbon dioxide reduction, J. Phys. Chem. Lett., 2017, 8 (1), 285–290*).

Thus, considering the high adsorption energy (-0.75 eV) of O_{ads} on Ag surface, it is reasonable that some amount of O_{ads} is present at the Ag surface under reductive potentials. This is an important topic that we will continue to investigate in future research.

4. The authors proposed a new CO₂ reduction reaction mechanism on Ag and supported it with computational modeling. What is the predicted Tafel slope based on the new mechanism? Is it consistent with previous experimental observations? Some discussions will be highly appreciated.

The focus of the current paper is on CO₂ adsorption and hydrogenation with no external applied potential.

We find that adsorption and activation for CO₂ on Ag is dramatically different than on Cu, which we expect is of significant interest to scientists studying the conversion of CO₂ to useful molecules. These new insights may suggest new strategies to optimize activity and selectivity.

As we advance to the point of applying external potential in the presence of electrolyte, we will certainly predict the Tafel slope and compare with previous experimental observations. To predict the performance, we will use Caltech's Grand Canonical methods to predict the free energy activation barriers as a function of applied potential, then do a microkinetics

analysis, and calculate the current as a function of applied potential. For example, see H. Xiao; H. Shin & W.A. Goddard III. Proc. Natl. Acad. Sci. U.S.A. 115 (23):5872–5877 (2018)

We have inserted the following sentences in the text as clarification for the readers (page 12, line 304-307):

“Future studies will include operando spectroscopic characterizations of these adsorbates under external potentials, and we will predict the Tafel slope to compare with previous experimental observations and to gain more insights into the new mechanism.”

NCOMMS-18-26430-T.

"Dramatic differences in CO₂ adsorption and initial steps of reduction between Ag and Cu"

Reviewer #2 (R2):

The manuscript describes computational and experimental work on CO₂ adsorption and reduction on Ag(111). The results are compared to previous results by the Authors on Cu(111). The experiments report AP-XPS of the C 1s binding energy while the computations have been performed within the density functional theory using a state-of-the-art code. The main conclusions are that H₂O strongly affects the activation and that the mechanisms on Cu and Ag are different.

There is currently considerable interest in the activation of CO₂ and the idea to compare different metal surfaces is interesting and important. This kind of work has the potential to enhance the understanding and ultimately aid the development of catalysts with favorable performance. The manuscript is interesting and should be considered for publication after the Authors have addressed the following three major points:

Response to Reviewer #2. Thank you for your helpful comments and suggestions. We in particular appreciate your strong support for publication in Nature Communications. Our response to each question is in blue bold face below.

1. A new mechanism is suggested for CO₂ adsorption on Ag(111). In particular, CO₂ is proposed to adsorb on oxygen covered Ag(111) by the formation of a charged O-CO₂ species. There are issues with this suggestion. **Adsorbed oxygen is at the applied conditions unstable with respect to the formation of "surface oxide" structures.** There are a range of different structures that may form depending on growth conditions. The p(4x4) is perhaps the mostly discussed structure in the literature. The Authors should consider CO₂ adsorption on such structures.

Author Reply: We thank the reviewer for these valuable suggestions. It is well known that the sticking probability of molecular oxygen is low (5×10^{-6}) on Ag. Thus, the formation of p(4x4)-O structure on Ag in previous studies was achieved by introducing high pressure O₂ (200 mbar or higher), or dosing atomic O (produced with a commercial thermal gas cracker), or dosing NO₂. (*Ref: Structure of Ag(111)-p(4x4)-O: no silver oxide, PRL 96, 146102 (2006); Carbonate formation on p(4x4)-O/Ag(111), PRB 84, 115430 (2011)*).

However, in our current work, we introduced molecular oxygen under relatively low pressure, which leads to a relatively low coverage of oxygen on Ag surface. Thus, we believe that O adsorbs (O_{ads}) on Ag three-fold sites instead of forming the p(4x4)-O structure. We have several reasons to support this proposal.

First, some recent studies found that the formation of p(4x4)-O structure on Ag(111) surface with O₂ adsorption occurs only at high surface oxygen coverage. It is clear that oxygen adsorbed on Ag forms some local disordered oxide phase in the initial stage, while

nucleation of $p(4\times 4)$ -O phases starts after saturation of this disordered phase. Moreover, these studies proposed that O sits on the Ag three-fold sites in this disordered phase, just as for our QM calculations. Additionally, the adsorption configuration of O on Ag showed exactly the same adsorption energy that we calculated in our work. (*Ref: Adsorption of O₂ on Ag(111): evidence of local oxide formation, PRL 117, 056101 (2016); Adsorption of molecular oxygen on the Ag(111) surface: A combined temperature programmed desorption and scanning tunneling microscopy study, JCP 148, 244702 (2018)*)

Second, as stated above, the stability of the proposed structure in our work was verified by QM predictions that show an adsorption free energy of -0.75 eV at our pressure and temperature.

Third, the previous work reported that Ag surface with the $p(4\times 4)$ -O structure showed additional signals at lower binding energy regions, which were attributed to Ag-O bonding. In our work, we observed only the metallic Ag signals on O-covered surfaces, as shown in Figure R2 (Supplementary Fig. 6a).

Figure R2: Ag3d spectra of pristine and oxygen-covered Ag surface before CO₂ adsorption. This is a part of Supplementary Fig. 6.

Finally, the formation of the O=CO₂^{δ-} species is a surface reaction process. We have limited amounts of O_{ads} on the surface with continuous CO₂ flow to the Ag surface, thus the O is

rapidly consumed and the adsorption of $\text{O}=\text{CO}_2^{\delta-}$ is not limited by the pre-dosed O structure.

To help improve the clarity of the surface oxygen structure based on the reviewer's comment, we have added some related references and added the conclusion to the supplementary materials:

“The maintenance of the metallic state of Ag and the low coverage of oxygen on the Ag further ruled out the formation of the Ag (111)-p(4 × 4)-O surface reconstruction.”

2. A second issue is that it has previously been suggested that CO adsorption on p(4x4) leads to facile carbonate formation, see Phys. Rev. B, 2011, 84, 115430. The C 1s binding energy reported for the carbonate in the PRB paper is the same as reported for the O-CO₂ in the present manuscript. It is unclear why the Authors rule out the formation of a carbonate. The Authors need to strengthen the arguments with respect to the existence of an O-CO₂ species. The presented experimental result (only C 1s binding energy) is not convincing. XPS spectra of Ag and O should make it clearer which species that is present and the state of the surface. Another possibility would be to perform IR-measurements.

Author Reply: The surface reaction between O_{ads} and CO_2 has long been of interest. As the reviewer pointed out, all previous studies assigned the feature to be CO_3 . They assumed it be an ionic carbonate on surface with an even charge distribution over the three O atoms with the 4+ valence state of the C atom. However, none of them identified the geometric configuration of CO_3 on surface.

There are previous studies showing that metal carbonate species have a C1s peak located at 288.4 eV or higher. Indeed Ag_2CO_3 shows a C1s peak at 288.6 eV (as shown in the Figure R3). (*Ref: Ion-exchange preparation for visible-light-driven photocatalyst AgBr/Ag₂CO₃ and its photocatalytic activity, RSC Adv., 2014,4, 9139-9147; Surface characterization study of the thermal decomposition of Ag₂CO₃ using X-ray photoelectron spectroscopy and electron energy loss spectroscopy, Journal of Electron Spectroscopy and Related Phenomena 107 (2000) 73–81*). We also have experimental data showing a small C1s peak located around 288.7 eV for $\text{CO}_2+\text{H}_2\text{O}$ co-adsorption (Supplementary Fig. 9, previously denoted as Supplementary Fig. 6), which is assigned to an ionic carbonate species.

Figure R3: The survey, Ag3d, C1s, and O1s spectra recorded on Ag_2CO_3 . The figure is adapted from RSC Adv., 2014, 4, 9139-9147

Owing to the new APXPS technique and the QM calculations, we have obtained new insights to challenge the previous assignment. We believe our comprehensive study provides a new assignment to the surface adsorption species with better accuracy. Specifically, we believe that the “carbonate” discussed in the PRB paper show a configuration similar to what we report in this current study. In our work, we have clear predictions for the charge distribution of the surface adsorbate and on the vibrational frequencies. Our QM finds that the C atom in this surface adsorbate has a charge of +1.26, which is significantly smaller than that (4+) in the ionic carbonate. This charge distribution is consistent with the binding energy difference we observe experimentally between the $\text{O}=\text{CO}_2^{\delta-}$ and ionic carbonate. Thus, our current work reveals the surface configuration of $\text{O}=\text{CO}_2^{\delta-}$ adsorption on Ag surface and also provides insights on how it alters the reaction pathway of CO_2RR on the Ag surface.

3. The results on Ag(111) are put in perspective by comparison with CO_2 adsorption on Cu(111). In the case of Cu(111), it is argued that CO_2 adsorption is stabilized by subsurface oxygen. However, this is a controversial and recent calculations indicate that subsurface oxygen is highly unstable and would convert to surface oxygen during typical reaction conditions. See Garza et al. J. Phys. Chem. Lett., 2018, 9, 601. The Authors need to discuss this point as subsurface oxygen appears to be unlikely.

Author Reply: We thank the reviewer for bringing up these important points. We are interested in the existence of both subsurface and surface oxygen on Cu surface. Our

previous QM calculations on Cu used the advanced M06 version of DFT theory that is optimized to describe both van der Waals attraction and reaction pathways, whereas Garza et al. used the semiempirically modified RPBE method for oxygen and the SCAN+rVV10 functional for physisorption of CO₂ with copper.

Our previous QM calculations were carried out at experimental conditions with gas phase CO₂ and H₂O (total pressure 0.7 torr, and room temperature), which is directly comparable to this current manuscript.

On the other hand, Garza et.al carried out the calculations with electrolyte and external potential, which is valuable but not directly comparable.

Our work calculated the free energy of binding of the various species showing the stability of the various species under the experimental pressures and temperatures. This led to excellent agreement with the APXPS chemical shifts. The experimental evidence of subsurface oxide is quite clear from the O1s spectra characterizations. Also, the experimental results of adding additional O experimentally confirmed our QM predictions.

Even so, there are some consistencies between Garza's work and our work. We both found that the *b*-CO₂ can only be stable with extra charge transferred to CO₂ to change the molecule structure. While the M06 DFT finds subsurface O changes the Cu valence state to provide extra charge, Garza applied an external potential that can provide extra charge that stabilized the bent configuration. Although our Cu experimental data does not include electrolyte and applied potential, our experiment together with the theory does show that extra charge can stabilize the *b*-CO₂ with H₂O.

Summarizing. The previous experiments prove the existence of subsurface O that the M06 DFT also finds and the QM and APXPS are fully consistent.

The fact that Garza does not find subsurface O to be stable may indicate that RBE is not accurate for such calculations or it may indicate the difference in having an external potential and electrolyte

With regards to this study we have inserted the following sentences in the section 2, Supplementary Materials as clarification for the readers (page 6, line 127-156):

"In a recent study performed by Garza et al., the stability of subsurface oxygen in Cu is questioned. Thus, we want to further clarify the Cu results by comparing the differences and consistencies between our previous work with Garza's.

We are interested in the existence of both subsurface and surface oxygen on Cu surface. Our previous QM calculations on Cu used the advanced M06 version of DFT theory that is optimized to describe both van der Waals attraction and reaction pathways, whereas Garza et

al. used the semiempirically modified PBE method for oxygen and the SCAN+rVV10 functional for physisorption of CO₂ with copper. Our previous QM calculations were carried at experimental condition with gas phase CO₂ and H₂O (total pressure 0.7 torr, and room temperature), which could be directly compared to this current manuscript. On the other hand, Garza et.al carried out the calculation with electrolyte and external potential, which is valuable but not directly comparable.

Our work calculated the free energy of binding of the various species showing the stability of the various species under the experimental pressures and temperatures. This led to excellent agreement with the APXPS chemical shifts. The experimental evidence of subsurface oxide is quite clear from the O1s spectra characterizations. Also, the experimental results of adding additional O experimentally confirmed our QM predictions.

Even so, there are some consistencies between Garza's work and our work. We both found that the b-CO₂ can only be stable with extra charge transferred to CO₂ to change the molecule structure. While the M06 DFT finds subsurface O changes the Cu valence state to provide extra charge, Garza applied an external potential that can provide extra charge that stabilized the bent configuration. Although our Cu experimental data does not include electrolyte and applied potential, our experiment together with the theory does show that extra charge can stabilize the b-CO₂ with H₂O.

Summarizing. The previous experiments prove the existence of subsurface O that the M06 DFT also finds and the QM and APXPS are fully consistent.”

NCOMMS-18-26430-T.

"Dramatic differences in CO₂ adsorption and initial steps of reduction between Ag and Cu"

Reviewer #3 (R3):

Response to Reviewer #3. Thank you for your helpful comments and suggestions, repeated below. Our response is in blue bold face.

Reviewer #3 (Remarks to the Author):

The authors report results of an experimental and computational study in which they claim that CO₂ adsorption on Ag produces a carbonic acid species of the form, O=CO₂d-, by coordinating with a chemisorbed O-atom, and that this species also forms in the presence of H₂O and hydrogen-bonds with co-adsorbed H₂O molecules. The authors present DFT results which predict that the carbonic acid species is stable and energetically-preferred over other configurations. They also present AP-XPS (ambient pressure X-ray photoelectron spectroscopy) results which they claim support their assignments. **I find that their AP-XPS spectra are not convincing in supporting the proposed assignments, and identify several issues that raise significant doubts about the interpretation of their experimental results.** The paper is also quite difficult to follow in my opinion. I note that the Supplementary Information is nearly twice as long as the main manuscript, and the authors repeatedly direct the reader to the SI for important information. At no point in the main manuscript or the SI do the authors clearly show how the experimental AP-XPS spectra **agree unambiguously with the DFT predictions.** Further, the reported C 1s spectra (SI, Figure 6) indicate that their polycrystalline Ag foil was **contaminated with several, unknown carbon species.** As such, assigning one of the several C 1s peaks to a O=CO₂d- species is highly questionable. This paper is unpublishable in its current form. More specific comments are as follows:

Author Reply: We thank the reviewer for sharing his/her thoughts on our paper. We apologize that the reviewer finds it hard to follow our paper. We answered all the questions raised by the reviewer, and changes are made in the revised manuscript. We believe this revision based on the reviewers' comments has improved our paper significantly. Our work demonstrates how experimental and theoretical analyses can be used synergistically to determine different aspects of the first step of activating CO₂ on Cu and Ag surfaces.

1) Experiments were performed using a polycrystalline Ag foil whereas the DFT calculations modelled a Ag(111) surface. The foil exposes many types of surface sites that are not reproduced by the Ag(111) model. As such, comparison between the experimental and computational results is questionable at the outset.

Author Reply: We believe correlating the experimental results on polycrystalline Ag foil with the theoretical simulation on Ag (111) surface is reasonable because the (111) surface is closest packed, energetically the most favorable, and the most abundant surface for fcc metals (such as Ag and Cu). To be specific, the surface energies order as (111) < (100) < (110). Using the most abundant surface facet in modeling polycrystalline metal is common

practice, which is the essential philosophy behind mean field theory: the study of a complex stochastic many-body-problem can be reduced to the study of one-body-problem where the effects of defects, facets with low populations are averaged out.

With regards to this study we have cited some references and inserted the following sentences in the texts as clarification for the readers (page 4, line 84-88):

“As the (111) surface is the closest packed, energetically the most favorable for fcc metals (such as Ag and Cu), experimental evidence indicated that silver (and Cu) catalyst treated with high temperature exposed this facets. Thus our simulations were performed based on the Ag (111) surface to correlate with the experimental observations on vacuum annealed polycrystalline Ag surface.”

2) The authors state that DFT predicts that the formation of subsurface oxygen in Ag(111) is energetically unfavorable, and they thus omit subsurface oxygen from further consideration in their study, including when they assign O 1s peaks in the experimental data. I note, however, that the formation of subsurface oxygen in Ag and its role in mediating surface chemistry on Ag has been reported by numerous authors for many years. The authors make no mention of the vast literature on this topic.

Author Reply: Contrary to the reviewer’s claim, the vast literature all shows that the barrier going from surface oxygen to subsurface oxygen on Ag is ~1eV. For example, Figure 5 of *PHYSICAL REVIEW B* 67, 045408 (2003) (as shown in Figure R4) clearly shows that the transition barrier from surface oxygen to subsurface oxygen on Ag(111) surface is +0.86eV, whereas the reverse barrier from subsurface to surface oxygen is only +0.18eV. Assuming a Boltzmann distribution, $F(\text{state2})/F(\text{state1}) = \exp[(E1-E2)/kT]$, using $E1-E2 = 0.68\text{eV}$, and kT at room temperature (experimental condition) = 0.02eV, the population of surface oxygen is around $\exp(34)=5.8 \times 10^{14}$, which is around 10^{14} times more than subsurface oxygen. Therefore, our omission of subsurface oxygen is quite justified. Subsurface oxygen is indeed energetically unfavorable under our experimental condition.

FIG. 5. Adsorption energy of oxygen at a coverage of 0.11 ML for penetration into the subsurface octa site from the on-surface fcc-hollow site through the first Ag layer, and from the octa site to the “bulk” tetrahedral site, through the second Ag layer (from right to left). From the tetra-I site, O will move to the more favorable bulk octa site under the second Ag layer. The short lines indicate the adsorption energy with respect to half of the binding energy of the oxygen molecule (experimental value). “TS” represents the transition states, and the numbers on the arrows indicate the energy barriers.

Figure R4 adapted from PHYSICAL REVIEW B 67, 045408 (2003)

With regards to this study we have added the following sentences in the Section 3 of Supplementary information as clarification for the readers (page 7, line 176-186):

“Previous studies included some discussion on the subsurface O in the Ag system, which is introduced through the grain boundary, defects in the structure, and diffusion of the surface oxygen into the bulk. These cases required moderate to high temperature and high oxygen coverage. Moreover, Li et al. performed a series of studies examining the stability of subsurface oxygen in Ag and found that the transition barrier from surface oxygen to subsurface oxygen on Ag(111) surface is +0.86eV, whereas the reverse barrier from subsurface to surface oxygen is only +0.18eV, leading to the population of surface oxygen is around $\exp(34)=5.8 \times 10^{14}$, which is around 10^{14} times more than subsurface oxygen. It is found that crystal expansion is needed to stable subsurface oxygen, where the high oxygen coverage is needed”

3) The absolute C 1s binding energies computed using DFT are on the order of 20 eV lower than experimental values. The authors claim that differences in the C 1s core levels that are less than 0.5 eV are accurate. I find this claim highly questionable, even though they cite a paper in the SI that they state supports the idea.

Author Reply: The major assumption in the theory is that the structure is not allowed to relax geometrically upon ionization, which leads to a constant correction on all C1s binding energies. Thus we need to compare relative energies, which agree well with experiment, rather than the absolute energy. *This procedure is described in page 5, line 128 to page 6,*

line 129: The $O=CO_2^{\delta-}$ C1s peak BE has been set as the reference point for subsequent experiments with H_2O .

4) The authors predict with DFT that the $O=CO_2^{\delta-}$ species features two distinct O-atoms, in a 2:1 ratio, that have different O 1s binding energies. However, they do not present experimental evidence to support this prediction.

Author Reply: We apologize that we did not make this clear in our paper. In fact, the O1s spectra for the O atoms do not provide much discrimination in these states, especially those with H_2O molecules attached. However, we agree that the O1s spectra for the case of CO_2 adsorption alone does provide insight to elucidate the surface chemistry. We have added the O1s spectra in the Supplementary Fig. 6. The two peaks that represent 2 O atoms attached to Ag surface and the single O atom in the $C=O$ bond, were used to fit the spectra. The energy difference between these two peaks was set as 0.7 eV based on the QM results. This leads to 2:1 peak intensity ratio. Thus the peak fitting of the experimental data supports the QM results.

Figure R5: O1s spectra of pristine and oxygen-covered Ag surface after CO_2 adsorption. The spectra recorded on pristine, low-oxygen covered, and high-oxygen covered Ag surface were represented in black, red, and blue, respectively. This is a part of Supplementary Fig. 6.

We thank the reviewer's suggestion to make this point clear.

With regards to this study we have inserted the following sentences in the texts as clarification for the readers (page 6, line 163- page 7, line 168):

“Moreover, the experimental O1s spectra shown in Supplementary Fig. 6 provide insight to elucidate the surface chemistry. The two peaks that represent 2 O atoms attached to Ag surface and the single O atom in the C=O bond, were used to fit the spectra. The energy difference between these two peaks was set as 0.7 eV based on the QM results. This leads to 2:1 peak intensity ratio. Thus the peak fitting of the experimental data supports the QM results.”

5) In the SI, the authors report O 1s spectra of clean and oxygen-exposed Ag foil and deconvolute the spectra into three peaks (528.5, 530.3, 531.5 eV). They assign these peaks to chemisorbed atomic oxygen, the O=CO₂^{δ-} species and OH species. **How can the peak at 530.3 eV originate from the O=CO₂^{δ-} species when no such species should be present?** This is illogical because the experiments were performed on O-exposed Ag, without the presence of CO₂, and actually suggest that the peak at 530.3 eV arises from a **surface carbon species** other than the O=CO₂^{δ-} species. Also, are they claiming that OH is a contaminant? Again, the O 1s spectra are reported for O-exposed Ag foil so OH could only be caused by contamination, which is clearly a possibility (see below). I note, however, that **subsurface O** in Ag has been reported to give a distinct O 1s peak in prior studies.

Author Reply: We want to further clarify the surface chemistry by combining the C1s and O1s spectra. As shown in the Figure R6, the pristine Ag surface prior to O₂ dosing did not show any peaks corresponding to C1s and O1s spectra, indicating that the Ag surface is clean with no detectable C- and O- containing contaminations.

With O₂ adsorption, we found two peaks showing up in the O1s spectrum. The lower binding energy peak (peak A) is assigned to the chemisorbed O on Ag, which is consistent with previous studies performed by Campbell group. More references have been added to support this assignment. The peak located at the high binding energy region (peak B), is the one we assign to O=CO₂^{δ-}. This assignment is supported by checking the C1s signal and the C:O atomic ratios, which are around 1:3 during the O₂ adsorption process. Since the peak position of this species in both the C1s and O1s spectra is located at identically the same position as those we observed later with CO₂ adsorption, we are confident to assign them to O=CO₂^{δ-}.

(Figure R6 (and the new Supplementary Figure 7): (a) O1s and (b) C1s spectra taken on Ag surface during O₂ dose. The spectra taken at UHV, 40 mTorr O₂ at room temperature, 40 mTorr O₂ at around 400 K, and 40 mTorr O₂ at 430K were recorded as black, red, blue, and pink, respectively, from bottom to top.)

As described in the Fig. 3 (previously denoted as Supplementary Fig. 7), it is clear that the adsorption of CO₂ on O-covered surface is very fast. Thus it is reasonable that the very small amount of residual CO₂ gas in chamber can react with surface O and form this O=CO₂^{δ-} species. It should also be noted that the existence of this small amount of CO₂ gas in the chamber is expected under an ambient pressure experiment, since adsorbed CO₂ gas on chamber wall can easily be replaced by O₂ during the O₂ dosing experiment. Also, we should emphasize that this peak can only be observed on O₂ covered Ag surface. The absence of this peak on the pristine Ag surface is consistent with our time dependent experiment results (as shown in Fig. 3 previously denoted as Supplementary Fig. 7).

Figure R7 (a repeat of Figure R1 above). O1s spectra of pristine and oxygen-covered Ag surfaces before and after CO₂ adsorption. This is a part of Supplementary Fig. 6.

We apologized that we did not add the scale bar in the previous version figure, which leads to some misunderstanding that can cause one to think the surface peak originated from O=CO₂^{δ-} species before CO₂ adsorption is high. To clarify this point, we have revised the previous figure and made the new figure, as shown in Fig. R7 (the new Supplementary Fig. 6), it is clear that the coverage of O=CO₂^{δ-} before CO₂ exposure is around 5% of the amount present after CO₂ exposure. Thus, the signals from reactions between residual CO₂ with O_{ads} on Ag did not influence our data analysis and conclusion.

We have assigned the O1s peak at the highest binding energy as OH. However, we need to point out that this peak is quite weak, with an intensity at noise level. We believe this small amount OH species does not influence the adsorption behavior of CO₂ on Ag and our data analysis process.

As we explained in the previous section, we conclude that there is no subsurface oxygen species in Ag, and our peak assignment is consistent with the prior studies.

With regards to this study we have inserted the following sentences in the texts of supplementary materials as clarification for the readers (page 16, line 332 -page 17, line 344):

“This assignment is supported by checking the C1s signals and the C:O atomic ratio, which are around 1:3 during the O₂ adsorption process (Supplementary Fig. 7). Since the peak position of this species in both the C1s and O1s spectra is located at the identical position as those we observed later with CO₂ adsorption, we are confident to assign them to O=CO₂^{δ-}. This

is further evidenced by its unstable of peak B above 430K. This is against the previous assigned bulk dissolved O peak, locating at similar position, is stable at up to 800K. By applying the sensitivity factors for both Ag3d and O1s, which are about 1.80 and 0.32, respectively, under photon energy of 670 eV, the Ag:O atomic ratio is around 0.01 and 0.015 for low and high oxygen covered surface, respectively. The maintenance of the metallic state of Ag and the low coverage of oxygen on the Ag further ruled out the formation of the Ag (111)-p(4 × 4)-O surface reconstruction.”

6) The authors assign a C 1s peak at 289.7 eV to the O=CO₂^{δ-} species on Ag foil. However, they do not present clear evidence to support this assignment. It is insufficient and unconvincing to assert that their assignment is correct only because DFT predicts that this species forms.

As far as I can tell, the only spectral evidence that they claim supports their peak assignment is that they can deconvolute the C 1s peak from (presumably) the O=CO₂^{δ-} species into three components when H₂O is present with CO₂. This is unconvincing for several reasons. A key reason is that they observe several other C 1s peaks as well, indicating that their Ag sample is highly contaminated with **unknown carbon-containing species**. Thus, the peak at 289.7 eV could arise from numerous species, not considered by DFT. An inability to correlate the C 1s feature with the O 1s feature raises further doubts about the assignment, though the **carbon contamination and polycrystalline surface** are much larger concerns.

Author Reply: We added the O1s spectra to the supplementary information (as shown in Supplementary Fig. 6(c)) to further support our assignment. Combining the C1s and O1s spectra, we found that the C:O ratio is close to 1:3 for all samples. Moreover, the C:O ratio of the adsorbate signals recorded during the time dependence experiment were calculated and proved to be around 1:3, as shown in Figure R8 the new Supplementary Fig. 10. This C:O ratio provides clear evidence to support the assignment of the adsorbate species as carbonate type species (1 carbon to 3 oxygen). Moreover, we investigated the charge distribution on this adsorbate, finding that the species is a chemisorbed carbonic acid-like species (O=CO₂^{δ-}), in which two O on the C bind to adjacent Ag bridging sites, with the third O forming a C=O double bond perpendicular to the surface. We further investigated the formation of O=CO₂^{δ-} by monitoring the effect of the pre-dosed oxygen (Supplementary Fig. 8, old Supplementary Fig. 5) and the dynamic process (Fig. 3, old supplementary Figure 7). All the information come together quite well.

Figure R8 (the new Supplementary Figure 10): The O:C ratio of the adsorbate signals recorded during the time dependence experiment

We also carried out extensive QM for all possible configurations of CO₂ adsorption on Ag, considering both physisorbed and chemisorbed CO₂ with and without surface/subsurface O. This work excludes the possible existence of all the other adsorption configurations, proving that O=CO₂^{δ-} is the only stable species on the Ag surface. These QM results are consistent with the experimentally observed single adsorbate peak.

We want to emphasize again that prior to CO₂ adsorption, the investigated Ag surface is clean with no detectable C- and O- containing contaminations (as shown in Figure R9, Supplementary Fig. 5). The other C1s peaks the reviewer suggested as unknown carbon contaminations are graphitic carbon that originates from CO₂ decomposition on the surface, which is part of the reaction step for CO₂ adsorption on clean Ag surface. This is further proved by the vanishing of these peaks when dosing CO₂ on an oxygen covered surface.

Figure R9: Survey, C1s, and O1s spectra of pristine Ag surface prior to the O₂ and/or CO₂ adsorption. This has been included in the Supplementary materials as Supplementary Fig. 5

Finally, as we explained in the first question, the QM results on Ag(111) agree well with the experimental observations.

7) The authors mention the CO₂ and H₂O partial pressures used in their experiments but do not mention the sample temperature during these exposures. They also mention that they vacuum annealed the Ag foil but do not mention the annealing temperature. Including such information is imperative and its omission is very surprising.

Author Reply: The sample is at 298K. We apologize that we did not make this information clearer. Besides recording this information in the supplementary information, we will make it clearer by adding related information to the main text. We also emphasized it again in the main text (Page 4, line 94-96):

“These and all other ΔG values are from QM calculations including zero point energy, entropy, and specific heat to obtain the ΔG at 298K and the pressure quoted.”

The sample preparation method have been recorded in the previous version:

Page 13, line 346 to line 349 in the main text, and page 5, line 117 to line 121 in the supplementary materials.

We also want to make it clearer that a clean surface was obtained by applying this method by inserting following sentence in the supplementary materials (Page 15, line 303 to 309):

“The Ag surface was characterized by XPS to ensure no detectable contamination on the surface. The survey with a binding energy range of -10 to 600 eV, and high resolution scans of C1s and O1s recorded at photon energy of 670 eV (Supplementary Fig. 5). The energy scale of the spectra was calibrated using the Ag3d_{5/2} peak locating at 368.2 eV. The survey spectra showed only Ag signals, including core level peaks and an auger peak. No detectable C- and O- based contamination were observed in the high resolution scans recorded in the insets.”

and main text (Page 6, line 142 to line 146):

“The adsorption states of CO₂ on various Ag surfaces were monitored by C1s APXPS. The pristine Ag surface showed no detectable carbon- and oxygen- based contamination (Supplementary Fig. 5), while dosing O₂ under different experimental conditions resulted in various oxygen coverages on Ag surface (Supplementary Fig. 6).”

8) In the SI, the authors report intensity ratios of the O 1s to Ag 3d peaks as a way to differentiate surfaces with differing amounts of oxygen. At the least the authors should apply sensitivity factors to account for differences in the O 1s and Ag 3d photoelectron cross sections. **However, a more appropriate way to quantify coverage is to collect spectra from a surface with a known coverage of adsorbate.** The current data does not provide a reliable estimate of the adsorbate amounts.

Author Reply: We apologize that we did not make this process clearer. We fully understand the method the reviewer proposes to calculate the surface adsorbate coverage may provide more accurate numbers. However, this method requires the application of STM or other techniques to clarify a “known coverage”.

Thus, in the previous version, we calculated the oxygen coverage by applying the sensitivity factors for both Ag3d and O1s, which are about 1.8 and 0.32, respectively, under a photon energy of 670 eV. We also considered that O adsorption on the Ag surface is sub-monolayer surface oxygen, while the probe depth for Ag is around 10 layers. We agree with the reviewer that this may not provide the most accurate value. Thus, we decided not to provide the detail value of O coverage, instead we provide now the Ag:O ratios (around 1:0.01 and 1:0.015) for both oxygen-covered surfaces and indicate which surface has the higher oxygen coverage. Although this information is not the main focus of this paper, providing this information will help to clarify the manuscript.

9) Supplementary Figure 6 shows C 1s spectra obtained from the Ag foil in the presence of CO₂. The C 1s spectra exhibit multiple peaks that are consistent with surface contamination by **adventitious carbon as well as various C-O moieties.** The corresponding intensities of these peaks are significant and in several cases larger than the peak that they claim arises from the O=C=O₂d- species. Thus, the authors are basing their interpretations about the formation of an (alleged) specific species on spectra obtained from a **highly carbon-contaminated, polycrystalline Ag foil.** Such an interpretation is highly questionable.

Author Reply: We want to emphasize again that prior to CO₂ adsorption the investigated Ag surface is clean with no detectable C- and O- containing contaminations as shown in Supplementary Fig. 5. It is a common practice to correlate the theoretical simulated results on Ag(111) (the most energy favorable facet among the Ag surfaces) with the experimental observed results on polycrystalline Ag surface.

We also have reorganized the paper to make it more readable, and all the revisions are highlighted by blue in the main text and supplementary materials.

Reviewers' comments:

Reviewer #1 (Remarks to the Author):

The authors provided additional results and explanations in the revision. Most of my concerns are addressed and therefore it could be considered for publication now.

Reviewer #2 (Remarks to the Author):

The Authors have responded to the issues raised in the previous round. Unfortunately, all my three major points remain. To make the manuscript publishable, the Authors need to clarify their response and add new calculations.

1. As the Authors indicate, it is possible that a disordered structure is formed on Ag(111) in the presence of low oxygen coverage. Moreover, as stated in the paper mentioned by the Authors (PRL 117, 056101) it is likely that oxygen induces surface vacancies also in the low coverage regime. The barriers for the formation of Ag vacancies are low as outlined in the recent work by Rocca and co-workers (PRB 98, 035405). Thus, the model considered by the Authors is probably idealized. Similar adsorption energies are not an argument for not considering CO₂ adsorption on models with Ag-vacancies - as the p(4x4) structure.

2. The Authors have added information on the O 1s binding energies. However, the figure captions are confusing. In the SI, Figure 6, b) is referred to as the C1s spectra. This is probably wrong. Both R1 and R5 are described as the O1s spectra before and after CO₂ adsorption. One of the columns in R1 indicates CO₂ adsorption. Which experiment is described in Figure R1a? Why is not the peak indicated as O-CO₂ deconvoluted in two peaks? In Figure R1b, why isn't there a Oad peak even though the oxygen coverage is indicated to increase?

The interpretation of the spectra as a O-CO₂ species instead of a carbonate is still not obvious. The Authors should present calculations where different configurations of CO₂ bonded to adsorbed O is presented. In particular, the Authors need to compare different vertical and horizontal configurations. I reiterate that an IR experiment probably would solve the issue.

3. The Authors should present numbers on the stability of subsurface oxygen in Cu(111) using some different density functionals. This is straight forward to do. The Authors state that subsurface oxygen provides extra charge to adsorbed CO₂. However, the intuitive picture is that subsurface oxygen takes charge from the surface atom, thus making it harder to transfer charge to the adsorbed molecule. The proposed simple mechanism is, thus, difficult to understand.

Reviewer #3 (Remarks to the Author):

The authors have provided more convincing arguments to support their interpretations, including showing that the O 1s peak is well described with two components in a 2:1 ratio separated by at 0.7 eV binding energy, consistent with the proposed CO₃ species.

Their interpretation of the O 1s peak assignments for the initially, clean Ag surface is also more clear. They added an intensity scale to illustrate that the O 1s peaks are much smaller before than after CO₂ exposure. Their logic for assigning the initial peak at 530.2 eV to the CO₃ species is now more convincing for a few reasons, including that its intensity is 3x larger than the C 1s peak, it appears at the same BE as that for the proposed CO₃ species and the O 1s and C 1s peaks diminish together with increasing temperature.

I agree that subsurface oxygen is unlikely to form at low coverage and 298 K (the temperature was initially omitted in the paper). As I mention above, their arguments about the 530.2 eV peak assignment are stronger now and convincing. However, researchers continue to present evidence for subsurface oxygen formation on Ag surfaces and the conditions under which this species forms are variable. As an example, reference 21 in the SI reports that subsurface oxygen forms even on close-packed Ag(111) at elevated O₂ pressure and 300 K, and that, after removal of the O₂ pressure, the subsurface oxygen concentration diminishes with increasing temperature above ~400 K or so. Those authors suggest that the subsurface O produces an O 1s peak at 530.2 eV. Again, I agree with the authors' interpretations after their clarification but this was by no means obvious in their initial paper, contrary to their suggestion in the rebuttal.

Use of Ag(111) to model polycrystalline Ag may be justified if the observed surface chemistry occurs on the majority facet(s). However, this assumption is questionable when adsorbate coverages are low because in this case the measured chemistry could be occurring predominantly, and even exclusively, at minority sites, including higher energy facets, e.g., (100), (110), and defects. The authors do not present quantitative information about the species coverages (O_{ad}, CO₃). I suggest that they address this issue, assuming that their data was obtained at high enough coverages to justify the assumption that they are observing chemistry/adsorption on the majority facet. For example, they report a O:Ag peak ratio of 0.01:1 and mention an approximate sampling depth of 10 Ag layers. Does this suggest a lower or upper bound O-coverage of 0.10 ML? The majority of the Ag XPS signal arises from the surface layer so the initial O-coverages may be lower than 0.1 ML – this is a relatively low coverage and could suggest chemistry dominated by minority sites. Again, my main point is that they can better justify use of the Ag(111) model and their interpretations of the XPS data if they can show that the adsorbate coverages probed with XPS are too high to be explained by minority-site chemistry.

The authors clarify that the initial Ag surface was clean and that the lower BE C 1s peaks from C=C and other carbon species are generated during the CO₂ exposure. I understood this from the outset and I apologize that I did not make my comments about these peaks more clear. The authors state that CO₂ dissociation on the Ag surface produces adsorbed O and various carbon species, particularly graphitic carbon. Can they cite papers demonstrating that CO₂ dissociation on Ag surfaces produces graphitic and/or atomic carbon? CO₂ dissociation is highly activated (as the authors know) and would typically produce CO + O on the surface, with the CO likely desorbing from Ag at 298 K. Their interpretation that graphitic carbon is a reaction product may indicate that adsorbed CO can undergo C-O bond cleavage. At least, the suggestion is that CO₂ decomposes to atomic and graphitic carbon on Ag. I find both possibilities interesting yet surprising, especially dissociation of adsorbed CO on Ag. Overall, the interpretation that CO₂ dissociation on Ag cleaves both C-O bonds, and produces graphitic/atomic carbon should be discussed further, particularly if such dissociation has not been previously observed.

An alternate possibility to CO₂ dissociation is that reactive carbon compounds (e.g., unsaturated hydrocarbons) are displaced from the chamber walls during the CO₂ exposures and that dissociation of such species rather than CO₂ produces the graphitic carbon and possibly other carbon species. Given that CO₂ dissociation is activated, its dissociation probability on Ag is probably quite low at 298 K so even a small quantity of "reactive" carbon contamination in the gas-phase could be sufficient to produce the observed low BE carbon species, e.g., if the CO₂ dissociation probability is 10⁻⁶ to 10⁻⁹, then reactive contaminants (O(1) sticking probability) at the ppm or ppb levels would adsorb at the same or higher rates as CO₂ dissociates. Can the authors rule out this possibility?

If the authors can prove that the graphitic carbon, etc. results from CO₂ dissociation and not contamination in the gaseous background of a multi-user chamber, generated after admitting 0.3 Torr of CO₂, can they make arguments that the presence of such species does not influence the proposed CO₃ species? Discussion of the species coverages may aid in developing such arguments. I note that the peaks from graphitic carbon are comparable in intensity to that from the proposed CO₃ species in

the data obtained during CO₂ exposure to clean Ag and CO₂ + H₂O exposures to Ag. On one hand, if the adsorbate coverages are moderate, thus better justifying use of the Ag(111) model, then their data was obtained from surfaces with a considerable quantity of dissociation products possibly in close proximity to the proposed CO₃ species. In such a case, omitting such species may or may not be justified in the modeling. On the other hand, if the adsorbate coverages are low, then it is unclear that Ag(111) is an appropriate model as the observed chemistry may be dominated by minority sites (e.g., defects, other facets).

NCOMMS-18-26430-T.

"Dramatic differences in CO₂ adsorption and initial steps of reduction between Ag and Cu"

Reviewer #1 (Remarks to the Author):

The authors provided additional results and explanations in the revision. Most of my concerns are addressed and therefore it could be considered for publication now.

Response to Reviewer #1. Thank you for your helpful comments and suggestions. We in particular appreciate your strong support for publication in Nature Communications.

NCOMMS-18-26430-T.

"Dramatic differences in CO₂ adsorption and initial steps of reduction between Ag and Cu"

Reviewer #2 (Remarks to the Author):

The Authors have responded to the issues raised in the previous round. Unfortunately, all my three major points remain. To make the manuscript publishable, the Authors need to clarify their response and add new calculations.

Response to Reviewer #2. Thank you for your helpful comments and suggestions. Our response to each question is in blue bold face below.

1. As the Authors indicate, it is possible that a disordered structure is formed on Ag(111) in the presence of low oxygen coverage. Moreover, as stated in the paper mentioned by the Authors (PRL 117, 056101) it is likely that oxygen induces surface vacancies also in the low coverage regime. The barriers for the formation of Ag vacancies are low as outlined in the recent work by Rocca and co-workers (PRB 98, 035405). Thus, the model considered by the Authors is probably idealized. Similar adsorption energies are not an argument for not considering CO₂ adsorption on models with Ag-vacancies - as the p(4x4) structure.

Author Reply: We thank the reviewer for these valuable suggestions. We have tested the structure suggested by the reviewer and investigated the adsorption of CO₂ on this surface. We found that after optimization there is no stable adsorption configuration.

We have included the results in the supplementary materials as clarification for the readers (page 10, line 233-page 11, line 256 in supplementary materials):

CO₂ adsorption on Ag surface with Ag vacancy

Some experimental studies have reported that O adsorption on Ag (111) surface induces the formation of Ag vacancies, which may act as active sites for CO₂ adsorption. We investigated the interaction of gas phase CO₂ with this oxygen-covered defective Ag surface. We examined all possible binding sites for forming the CO₃ structure. Prior to CO₂ adsorption we obtained a structure similar to that reported in previous work,¹ where 6 surface oxygen atoms surrounding each Ag vacancy (Supplementary Fig. 3). In the top view configuration, the first and second layer Ag atoms are highlighted by red and cyan outlines, respectively. As well established in the discussion above, the only stable configuration for CO₂ adsorption on Ag surface is the CO₃ structure having 2 oxygens bound to the three-fold site and 1 oxygen standing straight up and double bonded to the carbon. Thus, as labeled in Supplementary Fig. 3, three sites around each O are available for CO₂ attachment. Position 1 and 2 are found not to be possible for placing a CO₃ because of spatial constraints. From the side view, it is clear that the vacancy structure has Ag popping out in the Z direction, which creates a distortion that collides with the position of the C atom, making binding of CO₃ unfavorable for position 1 or 2. We attempted to put CO₂ at position 3 to form CO₃ structure as an initial structure (Supplementary Fig. 3). This structure is not stable

and relaxed to a linear CO_2 above a surface O, with $\Delta G = +0.44$ eV. We conclude that Ag vacancy surrounded with 6 oxygen atoms cannot act as an active site for CO_2 adsorption.

Supplementary Figure 3. The Ag vacancy in a 7×7 Ag(111) unit cell. (a) Side view and (b) top view of Ag(111) surface with 6 oxygen surrounding one Ag vacancy. (c) Side view and (d) top view of a starting CO_3 configuration on Ag(111) surface with 6 oxygen surrounding one Ag vacancy.

2. The Authors have added information on the O 1s binding energies. However, the figure captions are confusing. In the SI, Figure 6, b) is referred to as the C1s spectra. This is probably wrong. Both R1 and R5 are described as the O1s spectra before and after CO_2 adsorption.

Author Reply: We apologize for the mistake.

The Supplementary Fig. 8b (previously denoted as Fig. 6b) described the O1s spectra of clean and oxygen treated Ag surfaces before reactions.

Supplementary Figure 8. APXPS of pristine and oxygen-covered Ag surfaces and the adsorbates on them. a, Ag 3d spectra of pristine and oxygen treated Ag surface. The regions of the loss feature peaks was enlarged to indicate the metallic feature of the Ag surface after O₂ treatment. **b,** O 1s spectra of pristine and oxygen treated Ag surface. **c,** O 1s spectra of pristine and oxygen treated Ag surface after CO₂ adsorption.

One of the columns in R1 indicates CO₂ adsorption. Which experiment is described in Figure R1a?

The Figures R1a and R1b (also denoted as the Supplementary Fig. 8b and 6c (previously denoted as Fig. 6b and 6c)) describe the various Ag surfaces before and after CO₂ adsorption, respectively. The Figure R1a (also denoted as Supplementary Fig. 8b (previously denoted as Fig. 6b)) describes the clean and oxygen-covered surfaces before CO₂ adsorption. It has been described in the main text and supporting information.

We revised the related texts as clarification for the readers (page 6, line 151-153 in main text):

“..., while dosing O₂ under different experimental conditions resulted in various oxygen coverages on Ag surface (Supplementary Fig. 8), showing as changes of the O_{ad} peak intensity.”

And (page 18, line 363-368)

“The Ag 3d and O 1s spectra recorded on clean and oxygen covered surfaces are displayed in Supplementary Fig.8 a and b. Surfaces with low and high oxygen coverage surfaces were obtained by heating pristine Ag foil under 40 mTorr O₂ at 400 K for 5 mins and under 60 mTorr O₂ at 400 K for 15 mins, respectively. O 1s spectra recorded

on oxygen covered Ag surface showed three peaks locating at 528.5 eV, 530.3 eV, and 531.5 eV, respectively.”

Why is not the peak indicated as O-CO₂ deconvoluted in two peaks?

In the Supplementary Fig. 9 (previously denoted as Fig. 7), we provided the detailed evidence for the assignment for the peak located at 530.3 eV. The correlation between the O 1s signal and C 1s signal indicates that the peak located 530.3 eV corresponds to O=CO₂^{δ-}, while the peak located at 528.5 eV corresponds to surface Oxygen species without C.

We also provided detailed peak deconvolutions for the O=CO₂^{δ-} species, as shown in the Supplementary Fig. 8c (previously denoted as Fig. 6c), showing that the O 1s peak is well described with two components in a 2:1 ratio separated by at 0.7 eV binding energy, consistent with the QM derived O=CO₂^{δ-} species. To further clarify the peak origins, we added the detailed peak fitting for the O=CO₂^{δ-} signal in the Supplementary Fig. 8b.

In Figure R1b, why isn't there a O_{ads} peak even though the oxygen coverage is indicated to increase?

In the Figure R1b (also denoted as the Supplementary Fig. 8c (previously denoted as Fig. 6c)), the Ag surface after CO₂ adsorption is characterized with O 1s spectra. The O_{ads} peak that is visible before exposing to CO₂, disappeared after CO₂ adsorption. This is due to the surface reaction between O_{ads} and CO₂, which converted the surface O to form O=CO₂^{δ-} species on the surface. The disappearance of the O_{ads} peak is a strong indication for the appearance of this surface reaction.

The interpretation of the spectra as a O-CO₂ species instead of a carbonate is still not obvious. The Authors should present calculations where different configurations of CO₂ bonded to adsorbed O is presented. In particular, the Authors need to compare different vertical and horizontal configurations.

We thank the reviewer for this valuable suggestion. We have tried starting with several surface adsorption configurations, including vertical and horizontal configurations and further confirmed that the only stable species on Ag surface is O=CO₂^{δ-} with two oxygens on the surface and one C=O bond pointing up.

We revised the manuscript by adding related texts as clarification for the readers (page 6, line 138-line 146 in main text):

To advance the comprehensive understanding on the stability and properties of CO₃ structure on the Ag surface, we also investigated starting with vertical and horizontal CO₃ configurations on the Ag(111) surface (Supplementary Fig. 2). We found that the structure with one O bridging to the surface and two C-O bonds pointing up is not stable, with E_{ads} = +0.32 eV. This starting structure rotated to form the stable bidentate species. We also examined the stability of the horizontal CO₃ configuration with three C-O bonds constrained to be parallel to the Ag surface. This configuration is not stable.

The CO_2 bonding energy to form this horizontal structure is $\Delta E_{\text{ads}} = -0.34 \text{ eV}$, $\Delta G = +0.13 \text{ eV}$.

And (page 10, line 219-line 232 in supplementary materials):

CO₃ configurations on Ag surface

We have carried out several QM calculations for the configuration of CO_3 structure on Ag surface from CO_2 adsorption to a chemisorbed O atom. We found that the only stable CO_3 structure is with two O on the surface and one C=O bond perpendicular to the surface as shown in Figure 2. We found that positioning one O on the surface and two C-O bonds pointing to vacuum is not stable with an adsorption energy of $+0.32 \text{ eV}$ (Supplementary Fig. 2). Minimizing this monodentate structure leads to the bidentate structure. We also carried out QM calculations for the horizontal configuration with three C-O bonds parallel to the surface. This configuration is also not stable with an adsorption energy of -0.34 eV but $\Delta G = +0.13 \text{ eV}$. Minimizing this structure leads to the bidentate structure.

Supplementary Figure 2. The QM predictions of CO_3 configurations on the Ag surface. a, vertical CO_3 configuration on Ag surface with one O on the surface. b, The horizontal CO_3 configuration on Ag surface with three C-O bonds parallel to the surface.

I reiterate that an IR experiment probably would solve the issue.

We thank the reviewer for this valuable suggestion. IR experiments will be beneficial for obtaining additional understanding of the formation of the $\text{O}=\text{CO}_2^{\delta-}$ species from CO_2 adsorption. However, we do not have access to this kind of measurements due to the requirements of atomic level clean surfaces and *in-situ* ambient pressure measurements without exposing to any other gases excluding CO_2 . Adding this capability to our instrument is important and planned. We believe this will become possible in the future.

To advance the understanding of the interaction between CO₂ and Ag surface, we also provided the QM predicted vibration data for all the surface adsorbates in this study as a database to compare with future IR studies (Supplementary Table 2 (previously denoted as Table 1)). Our work has established the adsorption of CO₂ on the Ag surface showing the stable surface configuration to be O=CO₂^{δ-} where two O attach on the surface with one C=O bond perpendicular to the surface through comprehensive combination of QM studies and experimental characterization.

3. The Authors should present numbers on the stability of subsurface oxygen in Cu(111) using some different density functionals. This is straight forward to do. The Authors state that subsurface oxygen provides extra charge to adsorbed CO₂. However, the intuitive picture is that subsurface oxygen takes charge from the surface atom, thus making it harder to transfer charge to the adsorbed molecule. The proposed simple mechanism is, thus, difficult to understand.

Author Reply: We thank the reviewer for these valuable suggestions. We have tested the stability of the subsurface O in the Cu system using different functionals. The discussion and results are included in the Supplementary materials (page 6, line153-page 7, line 161):

We have also tested the stability of the subsurface O in the Cu system by checking two subsurface oxygen tetrahedron configurations: where configuration Cu-O_{sub-1} has subsurface oxygen above second layer Cu, and configuration Cu-O_{sub-2} has subsurface oxygen below second layer Cu. We carried out and compared using three different functional: PBE (GGA level), PBE_D3 (GGA level), and B3LYP (Hybrid functional), all of them show at least one stable structure for subsurface oxygen in Cu under standard conditions. The results are summarized in Supplementary Table 1.

Functional	Geometry	ΔE (eV)	ΔG (eV)	
			Standard condition	Stability
PBE_D3	Cu-O _{Sub-1}	-0.53	-0.18	Stable
	Cu-O _{Sub-2}	-0.18	0.17	Unstable
PBE	Cu-O _{Sub-1}	-0.37	-0.02	Stable
	Cu-O _{Sub-2}	-0.11	0.24	Unstable
B3LYP	Cu-O _{Sub-1}	-1.06	-0.71	Stable
	Cu-O _{Sub-2}	-0.91	-0.56	Stable

Supplementary Table 1: Stability of subsurface O in Cu system tested by different configurations and functions.

The chemical shift of these two geometries also compare well with the experimental values, as published in PNAS, 2017 114 (26) 6706-6711, which showed that subsurface oxygen further stabilizes *l*-CO₂ and *b*-CO₂.

We emphasize here that the difference between PNAS, 2017 114 (26) 6706-6711 and J. Phys. Chem. Lett. 2018, 9, 3, 601-606 are due to: 1) functional differences, which is tested above, and 2) condition differences, where Garza. et. al. considered a strong reducing potential and Favaro et. al. examined a neutral unbiased system.

Reviewer #3 (Remarks to the Author):

The authors have provided more convincing arguments to support their interpretations, including showing that the O 1s peak is well described with two components in a 2:1 ratio separated by at 0.7 eV binding energy, consistent with the proposed CO₃ species. Their interpretation of the O 1s peak assignments for the initially, clean Ag surface is also more clear. They added an intensity scale to illustrate that the O 1s peaks are much smaller before than after CO₂ exposure. Their logic for assigning the initial peak at 530.2 eV to the CO₃ species is now more convincing for a few reasons, including that its intensity is 3x larger than the C 1s peak, it appears at the same BE as that for the proposed CO₃ species and the O 1s and C 1s peaks diminish together with increasing temperature.

Response to Reviewer #3. Thank you for your helpful comments and suggestions, repeated below. We believe this revision based on the reviewers' comments has improved our paper significantly.

I agree that subsurface oxygen is unlikely to form at low coverage and 298 K (the temperature was initially omitted in the paper). As I mention above, their arguments about the 530.2 eV peak assignment are stronger now and convincing. However, researchers continue to present evidence for subsurface oxygen formation on Ag surfaces and the conditions under which this species forms are variable. As an example, reference 21 in the SI reports that subsurface oxygen forms even on close-packed Ag(111) at elevated O₂ pressure and 300 K, and that, after removal of the O₂ pressure, the subsurface oxygen concentration diminishes with increasing temperature above ~400 K or so. Those authors suggest that the subsurface O produces an O 1s peak at 530.2 eV. Again, I agree with the authors interpretations after their clarification but this was by no means obvious in their initial paper, contrary to their suggestion in the rebuttal.

Response We appreciate your comments on the improvement in our paper and that it is now clearer and more convincing than the previous version.

Use of Ag(111) to model polycrystalline Ag may be justified if the observed surface chemistry occurs on the majority facet(s). However, this assumption is questionable when adsorbate coverages are low because in this case the measured chemistry could be occurring predominantly, and even exclusively, at minority sites, including higher energy facets, e.g., (100), (110), and defects. The authors do not present quantitative information about the species coverages (O_{ad}, CO₃). I suggest that they address this issue, assuming that their data was obtained at high enough coverages to justify the assumption that they are observing chemistry/adsorption on the majority facet. For example, they report a O:Ag peak ratio of 0.01:1 and mention an approximate sampling depth of 10 Ag layers. Does this suggest a lower or upper bound O-coverage of 0.10 ML? The majority of the Ag XPS signal arises from the surface layer so the initial O-coverages may be lower than 0.1 ML – this is a relatively low coverage and could suggest chemistry dominated by minority sites. Again, my main point is that they can better justify use of the Ag(111) model and their interpretations of the XPS data if they

can show that the adsorbate coverages probed with XPS are too high to be explained by minority-site chemistry.

Author Reply: Thanks, We agree that adding surface adsorbate coverage aids in clarifying the surface chemistry that occurs at the surface majority sites and would further justify the use Ag(111) models. To exclude the possible influence from the surface sp^2 carbon signals, we investigated the CO_2 adsorption on Ag surface with the highest O coverage prior to CO_2 adsorption. After applying the sensitivity factor for Ag 3d and O 1s at 670 eV, the O:Ag atomic ratio is calculated to be around 0.2:1. Considering the depth profiles, which is around 10 layers, the $O=CO_2^{\delta-}:Ag_{surf}$ ratio is around 0.71. This high coverage of the $O=CO_2^{\delta-}$ adsorbate on the surface indicates that the surface chemistry happens on the majority sites. Thus the use of Ag(111) model is well justified.

We revised the manuscript by adding related texts as clarification for the readers (page 8, line 212-line 216 in main text):

Lastly, we made a rough estimate of the surface coverage, by calculating the Ag and O atomic ratio, and the $O=CO_2^{\delta-}:Ag_{surf}$ ratios, which are found to be around 0.4:1, 0.6:1, and 0.7:1. This indicates that the reaction between surface O and Ag to form $O=CO_2^{\delta-}$ happens at surface majority sites, justifying the use of the Ag(111) model in this study.

The authors clarify that the initial Ag surface was clean and that the lower BE C 1s peaks from C=C and other carbon species are generated during the CO_2 exposure. I understood this from the outset and I apologize that I did not make my comments about these peaks more clear. The authors state that CO_2 dissociation on the Ag surface produces adsorbed O and various carbon species, particularly graphitic carbon. Can they cite papers demonstrating that CO_2 dissociation on Ag surfaces produces graphitic and/or atomic carbon? CO_2 dissociation is highly activated (as the authors know) and would typically produce CO + O on the surface, with the CO likely desorbing from Ag at 298 K. Their interpretation that graphitic carbon is a reaction product may indicate that adsorbed CO can undergo C-O bond cleavage. At least, the suggestion is that CO_2 decomposes to atomic and graphitic carbon on Ag. I find both possibilities interesting yet surprising, especially dissociation of adsorbed CO on Ag.

Overall, the interpretation that CO_2 dissociation on Ag cleaves both C-O bonds, and produces graphitic/atomic carbon should be discussed further, particularly if such dissociation has not been previously observed.

An alternate possibility to CO_2 dissociation is that reactive carbon compounds (e.g., unsaturated hydrocarbons) are displaced from the chamber walls during the CO_2 exposures and that dissociation of such species rather than CO_2 produces the graphitic carbon and possibly other carbon species. Given that CO_2 dissociation is activated, its dissociation probability on Ag is probably quite low at 298 K so even a small quantity of “reactive” carbon contamination in the gas-phase could be sufficient to produce the observed low BE carbon species, e.g., if the CO_2 dissociation probability is 10^{-6} to 10^{-9} , then reactive contaminants (O(1) sticking probability) at the ppm or ppb levels would

adsorb at the same or higher rates as CO₂ dissociates. Can the authors rule out this possibility?

If the authors can prove that the graphitic carbon, etc. results from CO₂ dissociation and not contamination in the gaseous background of a multi-user chamber, generated after admitting 0.3 Torr of CO₂, can they make arguments that the presence of such species does not influence the proposed CO₃ species? Discussion of the species coverages may aid in developing such arguments. I note that the peaks from graphitic carbon are comparable in intensity to that from the proposed CO₃ species in the data obtained during CO₂ exposure to clean Ag and CO₂ + H₂O exposures to Ag. On one hand, if the adsorbate coverages are moderate, thus better justifying use of the Ag(111) model, then their data was obtained from surfaces with a considerable quantity of dissociation products possibly in close proximity to the proposed CO₃ species. In such a case, omitting such species may or may not be justified in the modeling. On the other hand, if the adsorbate coverages are low, then it is unclear that Ag(111) is an appropriate model as the observed chemistry may be dominated by minority sites (e.g., defects, other facets).

Author Reply: Thanks for the reviewer's valuable comments. We agree with the reviewer that it is hard for CO₂/CO to dissociate on Ag surface to form surface carbon. Indeed, we do not have direct evidence for the CO₂ dissociation on the Ag surface at ambient pressure. We agree with the reviewer that this claim is too strong. We agree with the reviewer that some small quantity of reactive carbon contamination gases may cause this low BE carbon peak, especially considering that the measurements were under ambient pressure. We also must mention that the formation of the low BE sp² carbon is associated with the timeframe of the O=CO₂^{δ-} species formation on the surface. The formation of O=CO₂^{δ-} is a slow process on clean Ag surface, leading to the Ag catalyst surface exposed to possible existing reactive carbon contamination gases. Moreover, on oxygen covered surface, the immediately formed O=CO₂^{δ-} layer further blocked the interaction between the Ag surface with possible existing reactive carbon contamination gases. Thus, sp² carbon species are only observed in the clean surface when exposed to CO₂ gases.

We believe that the formation of the sp² carbon did not influence the formation of the O=CO₂^{δ-} species, which is from the surface reaction between O and CO₂. First, the surface adsorbate C 1s peak intensity indicates that even with the lowest adsorbate coverage, corresponding to CO₂ interacting with the clean Ag surface, the surface coverage is around 0.4ML. Thus, the surface adsorbate coverage is high enough to ensure that the formation of O=CO₂^{δ-} species happens at the surface majority sites. Second, we found that CO₂ adsorption on oxygen covered Ag surface or co-dosed with O₂ on clean Ag surfaces gave rise to a single adsorbate C 1s peak. This peak showed exactly the same peak position and full width at half maximum compared to that observed on clean Ag surface. This is a strong evidence that the same reaction happens for the two surfaces, one with the sp² carbon formation and one without.

Summarizing, the surface adsorbates on Ag surface have moderate to high coverage that ensure the formation of $O=CO_2^{\delta-}$ species happened at the majority sites. Thus, the use of the Ag(111) is well justified.

We have inserted the following sentences in the text as clarification for the readers (page 6, line 159-161):

Low binding energy region from 282 eV to 286 eV represents the surface reaction products from possible reactive carbon compounds (e.g., unsaturated hydrocarbons) from the chamber.

And (page 8, line 194 to line 211)

During this dynamic process, the O:C atomic ratio were calculated to be around 3:1, validating the surface adsorbate of $CO_3^{\delta-}$ structure, as shown in Supplementary Fig. 12. The largely accelerated process for the surface to reach equilibrium by adding O_2 is due to the formation of surface oxygen. Since CO_2 adsorption on clean (non-oxygen pretreated) Ag surface requires a CO_2 dissociation process prior to the formation of the final surface adsorbate, the dynamics of $O=CO_2^{\delta-}$ formation on clean Ag surface is slower than that with the oxygen co-dosed.

It is well known that during ambient pressure exposure of CO_2 , possible residential reactive carbon compounds (e.g., unsaturated hydrocarbons) can be desorbed from the chamber. Thus, due to the slow surface reaction of CO_2 on the clean Ag surface could lead to a larger possibility for the Ag surface to be exposed to unsaturated hydrocarbons that can lead to the formation of the sp^2 carbon species. When the surface initially possesses surface O_{ad} (Supplementary Fig. 8) or co-dosed with O_2 (Fig. 3), CO_2 can directly adsorb on the surface to form $O=CO_2^{\delta-}$. This suppresses surface carbon formation as evident in the decrease of the surface carbon (mainly the sp^2 $C=C^{2-5}$) C 1s signals (Supplementary Fig. 11), resulting in more available surfaces sites to increase the amount of adsorbed $O=CO_2^{\delta-}$ (Fig. 3 and Supplementary Fig. 10).

References:

- 1 Andryushechkin, B. V. et al. Adsorption of O_2 on Ag(111): Evidence of Local Oxide Formation. *Physical Review Letters* **117**, 056101 (2016).

Reviewers' comments:

Reviewer #2 (Remarks to the Author):

The Authors have clarified several points in the previous versions of the manuscript. There is, however, still one important point to make clear, namely the occurrence of subsurface oxygen on Cu(111).

In the SI, the Authors state that they cannot make a one-to-one comparison with the results of Garza et al. I do not understand this statement. Garza et al. [Ref 15] used PBE and showed that subsurface oxygen is highly unstable with respect to oxygen on the surface. Figure 1 in Ref. 15. The black line is without potential. The used functional in Figure 1 is not a "semiempirical modified PBE method" as stated by the Authors. That subsurface oxygen is unstable with respect to oxygen on the surface (by 1.5 eV) makes the statements in Table S1 questionable. It is stated that the configurations are stable. With respect to what are those configurations stable? There is perhaps a stability wrt the gas-phase but not wrt to the surface site.

A minor point is that the Authors state in the abstract that converting CO₂ to fuels is "a national priority". It could be good to know which nation they are referring to. I would argue that the topic is of world-wide priority.

Reviewer #3 (Remarks to the Author):

The authors have generally addressed my concerns about their data and interpretations. However, they offer several rationalizations that could raise doubts, and that required two rounds of thorough review to finally reach. I remain somewhat uncertain about the validity of their claims. Nonetheless, their various explanations and justifications may be sufficient to support their interpretations, and the article is likely suitable for publication.

Note: The term "residential" should be replaced by "residual" in the new text that they added.

NCOMMS-18-26430-T.

"Dramatic differences in CO₂ adsorption and initial steps of reduction between Ag and Cu"

Dear Editors and Reviewers,

Thank you for the careful evaluations of our manuscript entitled "Dramatic differences in CO₂ adsorption and initial steps of reduction between Ag and Cu". All the comments are valuable and very helpful for revising and improving our paper. We have addressed each comment carefully and accordingly revised the manuscript. All the changes in the paper are highlighted in green. The response to the reviewer's comments can be found below:

Reviewers' comments:

Reviewer #2 (Remarks to the Author):

The Authors have clarified several points in the previous versions of the manuscript. There is, however, still one important point to make clear, namely the occurrence of subsurface oxygen on Cu(111).

Response to Reviewer #2. Thank you for your helpful comments and suggestions.

In the SI, the Authors state that they cannot make a one-to-one comparison with the results of Garza et al. I do not understand this statement. Garza et al. [Ref 15] used PBE and showed that subsurface oxygen is highly unstable with respect to oxygen on the surface. Figure 1 in Ref. 15. The black line is without potential. The used functional in Figure 1 is not a “semiempirical modified PBE method” as stated by the Authors. That subsurface oxygen is unstable with respect to oxygen on the surface (by 1.5 eV) makes the statements in Table S1 questionable. It is stated that the configurations are stable. With respect to what are those configurations stable? There is perhaps a stability wrt the gas-phase but not wrt to the surface site.

Author Reply: We thank the reviewer for these valuable comments. The reviewer is concerned about the stability of subsurface O for the Cu(111) surface. We do not believe that there is any issue here.

Xiao, Cheng, and Goddard (*Proceedings of the National Academy of Sciences* 114 (26), 6706-6711) showed that high level M06 DFT calculations for Cu(111) leads surface and subsurface O atoms stable with respect to O₂ gas. These results were confirmed with APXPS experiments reported in the same paper.

Similarly, Garza, Bell, and Head-Gordon (*J. Phys. Chem. Lett.*, 2018, 9 (3), pp 601–606) use a somewhat less accurate DFT (PBE without D3 corrections) to arrive at the same conclusion.

The quoted statements below can be found at Page 604, *J. Phys. Chem. Lett.*, 2018, 9 (3), pp 601–606:

“The present study also demonstrates that the oxidation of Cu(111) by O₂ to form surface and subsurface oxygen is strongly favored thermodynamically.”

and

“Thus, even at extremely low O₂ partial pressures ($\approx 1 \times 10^{-72}$ Torr at 298 K), oxygen capture at surface and subsurface sites is thermodynamically favored. The very high

sensitivity of reduced Cu to oxidation is also confirmed by the observation of subsurface oxygen in ultrahigh vacuum studies at pressures as low as 5×10^{-8} Torr. Hence, we suggest that it is virtually impossible to preclude the oxidation of fully reduced Cu by trace amounts of oxygen present in water once Cu is no longer under a reducing potential. We also note that O₂ sorption and the instability of subsurface oxygen relative surface oxygen that we report here are in agreement with previous theoretical studies of surface oxides on Cu(111) and with the recently reported work carried out using ¹⁸O labeled water discussed earlier in the text.”

Thus, we can ensure Review #2 that there is no discrepancy to be concerned with between the two works.

Additionally, we would like to direct Review #2 to Supplementary Table 1 where we indicate that all of the free energy values are referenced to O₂ gas-phase (standard temperature and pressure conditions).

A minor point is that the Authors state in the abstract that converting CO₂ to fuels is “a national priority”. It could be good to know which nation they are referring to. I would argue that the topic is of world-wide priority.

Author Reply: We thank the reviewer for these valuable suggestions. We have revised the text according to the reviewer’s suggestions (line 26, page 2):

Converting carbon dioxide (CO₂) into liquid fuels and synthesis gas is a world-wide priority.

Reviewer #3 (Remarks to the Author):

The authors have generally addressed my concerns about their data and interpretations. However, they offer several rationalizations that could raise doubts, and that required two rounds of thorough review to finally reach. I remain somewhat uncertain about the validity of their claims. Nonetheless, their various explanations and justifications may be sufficient to support their interpretations, and the article is likely suitable for publication.

Response to Reviewer #3. Thank you for your helpful comments and suggestions. We believe the revisions made basing on the reviewers' comments have improved our paper significantly. We in particular appreciate your strong support for publication in Nature Communications.

Note: The term "residential" should be replaced by "residual" in the new text that they added.

Author Reply: We are sorry for the typo, we have revised the text according to the reviewer's suggestions (line 202-204, page 8):

It is well known that during ambient pressure exposure of CO₂, possible residual reactive carbon compounds (e.g., unsaturated hydrocarbons) can be desorbed from the chamber.

Reviewers' comments:

Reviewer #2 (Remarks to the Author):

The Authors have responded to the previous comment. The issues, however, remain.

1) The Authors do not clearly state that subsurface oxygen for Cu(111) is unstable with respect to surface oxygen at moderate coverages. Subsurface oxygen has a negative Delta G with respect to gas-phase oxygen but not with respect to surface oxygen. This is what the is written in the last four lines in the cited part from the work by Garza et al. This should be clearly stated in the manuscript and supporting information.

The instability of subsurface oxygen with respect to surface oxygen has a coverage dependence as pointed out in PRB, 73, 165424 (2006). I cannot find any description of the coverage used in the calculations in Table 1 of the SI.

2) The functional used by Garza et al. is PBE. In the SI, the Authors still refer to this functional as "semiempirically modified PBE", which is not correct.

NCOMMS-18-26430-T.

"Dramatic differences in CO₂ adsorption and initial steps of reduction between Ag and Cu"

Dear Editors and Reviewers,

Thank you for the careful evaluations of our manuscript entitled "Dramatic differences in CO₂ adsorption and initial steps of reduction between Ag and Cu". All the comments are valuable and very helpful for revising and improving our paper. We have addressed each comment carefully and accordingly revised the manuscript. All the changes in the paper are highlighted in pink. The response to the reviewer's comments can be found below:

Reviewers' comments:

Reviewer #2 (Remarks to the Author):

The Authors have responded to the previous comment. The issues, however, remain.

Response to Reviewer #2. Thank you for your helpful comments and suggestions.

1) The Authors do not clearly state that subsurface oxygen for Cu(111) is unstable with respect to surface oxygen at moderate coverages. Subsurface oxygen has a negative Delta G with respect to gas-phase oxygen but not with respect to surface oxygen. This is what the is written in the last four lines in the cited part from the work by Garza et al. This should be clearly stated in the manuscript and supporting information.

The instability of subsurface oxygen with respect to surface oxygen has a coverage dependence as pointed out in PRB, 73, 165424 (2006). I cannot find any description of the coverage used in the calculations in Table 1 of the SI.

Author Reply: We want to clarify that our ΔG values are with respect to the gas-phase O_2 (standard conditions), which has been identified in the table and SI. This is because we are reporting a comprehensive study of first step activation of CO_2 on catalyst surfaces with various O_2 pre-dosing conditions.

It is downhill in free energy to form subsurface oxygen with respect to gas phase oxygen, so there is no questions regarding its stability in the system.

However, subsurface O is less stable than surface O on Cu, as reported by Garza. Indeed Graza et al. also agree that subsurface O is stable, but that subsurface O is less stable than surface O. Thus to quote Garza: *"The present study also demonstrates that the oxidation of Cu(111) by O_2 to form surface and subsurface oxygen is strongly favored thermodynamically. The free energies of sorption per oxygen atom ΔG_{sorp} estimated with the SCAN+rVV10 functional are -5.93 , -4.55 , and -4.03 eV for the surface, O_{hl} , and O_{hll} sites, respectively, in reasonable agreement the experimentally measured value of -4.47 eV. Thus, even at extremely low O_2 partial pressures ($\approx 1 \times 10^{-72}$ Torr at 298 K), oxygen capture at surface and subsurface sites is thermodynamically favored. The very high sensitivity of reduced Cu to oxidation is also confirmed by the observation of subsurface oxygen in ultrahigh vacuum studies at pressures as low as 5×10^{-8} Torr."*

We also want to emphasize that there are many experimental results that support the existence of subsurface O in the Cu(111), here we list a few: Detection of subsurface oxygen on Cu(111): correlation of second-harmonic generation and Auger electron spectroscopy observations, *Surface Science*, 1991, 257, 328-334; Kinetics of oxygen adsorption, absorption, and desorption on the Cu (111) surface, *J. Chem. Phys.* 1993, 98, 9167; Nature and distribution of

stable subsurface oxygen in copper electrodes during electrochemical CO₂ Reduction, *J. Phys. Chem. C*, 2017, 121, 25003–25009; Subsurface Oxygen in oxide-derived copper electrocatalysts for carbon dioxide reduction, *J. Phys. Chem. Lett.*, 2017, 8, 285–290.

To clarify these issues we carried out additional simulations and added the following paragraph to the Supplementary Materials, pointing out that subsurface O is stable on Cu relative to gas phase O₂, but it is less stable than surface O. The coverage we used for the surface and subsurface O stability simulation is 1/4 ML to get direct comparison.

This appears in (Supplementary Materials)

Page 6, line 125-line128):

The stability of subsurface oxygen in Cu was questioned recently in study performed by Garza et al.¹. This may have caused confusion in the community, so we want to clarify the Cu results for O atoms on Cu surfaces in a vacuum, comparing the differences and consistencies between our previous works with Garza's.

and Page 6, line 153-Page 8, line176):

*For the Cu(111) and Ag(111) surfaces, we examined the stability of surface and subsurface O. For Ag, subsurface O is not stable and transfers to form surface O without an energy barrier. For Cu(111) the DFT predicted energies for O atom at various positions on and in the Cu(111) surface (Supplementary **Fig. 1**) are summarized in Supplementary **Table 1**. O atom on the surface is bound by 2.53 eV with respect to 1/2 O₂ (gas phase) while subsurface O is bound by 0.83 eV in the tetrahedral site and by 1.18 eV in the octahedral site. These results are in line with the results by Garza et al.¹ Thus, both works reached the same conclusion that formation of subsurface O in Cu is strongly favored thermodynamically compared to gas phase O₂, but subsurface O is less stable than surface O. The appearance of subsurface O in the Cu is also further evidenced by many experimental studies.²⁻⁵*

Supplementary Figure 1: The subsurface and surface O in Cu system. The configurations represent (a) the octahedron subsurface oxygen, (b) tetrahedron subsurface oxygen (O below 3 fold site), (c) tetrahedron subsurface oxygen (O below top site), (d) fcc surface oxygen, and (e) hcp surface oxygen, respectively.

Structure	PBE		PBE-D3		Garza et.al	
	Energy (eV)	ΔE (eV)	Energy (eV)	ΔE (eV)	PBE (eV)*	(SCAN+rVV1) (eV)**
O atom	-1.68	N/A	-1.68	N/A	N/A	N/A
O-O bond	-6.50	N/A	-6.50	N/A	N/A	N/A
O-O bond (exp)	-5.16	N/A	-5.16	N/A	N/A	N/A
O₂ molecule (exp)	-8.52	N/A	-8.52	N/A	N/A	N/A
O₂ molecule	-9.86	N/A	-9.87	N/A	N/A	N/A
Cu	-55.27	N/A	-62.24	N/A	N/A	N/A
(a) O_{sub, octa}	-60.62	-1.08	-67.68	-1.18	-0.30	-4.55
(b) O_{sub, tetra, 3 fold}	-60.36	-0.83	-67.33	-0.83	0	-4.03
(c) O_{sub, tetra, top}	unstable	N/A	unstable	N/A	N/A	N/A
(d) O_{surf, fcc}	-61.85	-2.32	-69.03	-2.53	-1.84	-5.93
(e) O_{surf, hcp}	-61.72	-2.19	-68.89	-2.39	N/A	N/A
O_{Third layer, octa}	-60.01	-0.48	-66.96	-0.46	N/A	N/A

*reference to subsurface O at tetrahedron site (O below 3-fold site)

**reference to atomic O

Table S1. DFT predicted energies for O atom at various positions on and in the Cu(111) surface. The configuration of O_{sub, octa}, O_{sub, tetra, 3 fold}, O_{sub, tetra, top}, O_{surf, fcc}, and O_{surf, hcp} are displayed in the **Supplementary Fig. 1 (a)-(e)**, respectively. Because DFT does not describe the O-O bond strength accurately, we define the energy of the O atom species relative to O₂ as $\Delta E = E(\text{surface species and surface}) - E(\text{surface}) - \frac{1}{2} E(\text{O}_2 \text{ molecule from experiment})$. The coverage of oxygen used for the simulation is 1/4 ML.

References:

- 1 Garza, A. J., Bell, A. T. & Head-Gordon, M. Is Subsurface Oxygen Necessary for the Electrochemical Reduction of CO₂ on Copper? *The Journal of Physical Chemistry Letters* **9**, 601-606 (2018).
- 2 Bloch, J., Bottomley, D. J., Janz, S. & van Driel, H. M. Detection of Subsurface Oxygen on Cu(111): Correlation of Second-Harmonic Generation and Auger Electron Spectroscopy Observations. *Surface Science* **257**, 328-334 (1991).
- 3 Cavalca, F. et al. Nature and Distribution of Stable Subsurface Oxygen in Copper Electrodes During Electrochemical CO₂ Reduction. *The Journal of Physical Chemistry C* **121**, 25003-25009 (2017).
- 4 Eilert, A. et al. Subsurface Oxygen in Oxide-Derived Copper Electrocatalysts for Carbon Dioxide Reduction. *The Journal of Physical Chemistry Letters* **8**, 285-290 (2017).
- 5 Bloch, J. et al. Kinetics of Oxygen Adsorption, Absorption, and Desorption on the Cu(111) Surface. *The Journal of Chemical Physics* **98**, 9167-9176 (1993).

2) The functional used by Garza et al. is PBE. In the SI, the Authors still refer to this functional as “semiempirically modified PBE”, which is not correct.

Author Reply: Sorry for the mistake. We have revised the text according to the reviewer’s suggestions. It shows: (line 132, page 6):

....,whereas Garza et al. used the PBE method for oxygen and the SCAN+rVV10 functional for physisorption of CO₂ with copper.

REVIEWERS' COMMENTS:

Reviewer #4 (Remarks to the Author):

I have examined the sequence of responses to reviewer #2. Focusing on the last review/response exchange - in my opinion the authors have addressed the concern of the reviewer. They have provided the free energy of subsurface versus surface O and this was the main request by the reviewer. They have also addressed some of the other minor requests.

I will note that the free energy favors surface O over subsurface O, but the authors have done these calculations at low coverage (1/4 ML). As is noted in the reviewer comment/response - the favorability of the subsurface O will depend on coverage and temperature (which in turn depend on the experimental conditions). There may also be kinetic considerations in the presence of O in the subsurface in the experiments. Overall, there is sufficient data provided now to show the reader that subsurface O is more stable than gas phase O₂ but not as stable as surface O. Likely more detailed studies correlating experimental conditions to O on the Cu surface is needed to fully understand how subsurface O forms and this would be outside the scope of this present study.

NCOMMS-18-26430-T.

“Dramatic differences in CO₂ adsorption and initial steps of reduction between Ag and Cu”

REVIEWERS' COMMENTS:

Reviewer #4 (Remarks to the Author): I have examined the sequence of responses to reviewer #2. Focusing on the last review/response exchange - in my opinion the authors have addressed the concern of the reviewer. They have provided the free energy of subsurface versus surface O and this was the main request by the reviewer. They have also addressed some of the other minor requests. I will note that the free energy favors surface O over subsurface O, but the authors have done these calculations at low coverage (1/4 ML). As is noted in the reviewer comment/response - the favorability of the subsurface O will depend on coverage and temperature (which in turn depend on the experimental conditions). There may also be kinetic considerations in the presence of O in the subsurface in the experiments. Overall, there is sufficient data provided now to show the reader that subsurface O is more stable than gas phase O₂ but not as stable as surface O. Likely more detailed studies correlating experimental conditions to O on the Cu surface is needed to fully understand how subsurface O forms and this would be outside the scope of this present study.

Response to Reviewers:

Dear Editors and Reviewers,

Thank you for the careful evaluations of our manuscript entitled “Dramatic differences in CO₂ adsorption and initial steps of reduction between Ag and Cu”. We are glad we were able to answer all the questions and are grateful to the referee for their appreciation of this work.

REVIEWERS' COMMENTS:

Reviewer #4 (Remarks to the Author): I have examined the sequence of responses to reviewer #2. Focusing on the last review/response exchange - in my opinion the authors have addressed the concern of the reviewer. They have provided the free energy of subsurface versus surface O and this was the main request by the reviewer. They have also addressed some of the other minor requests. I will note that the free energy favors surface O over subsurface O, but the authors have done these calculations at low coverage (1/4 ML). As is noted in the reviewer comment/response - the favorability of the subsurface O will depend on coverage and temperature (which in turn depend on the experimental conditions). There may also be kinetic considerations in the presence of O in the subsurface in the experiments. Overall, there is sufficient data provided now to show the reader that subsurface O is more stable than gas phase O₂ but not as stable as surface O. Likely more detailed studies correlating experimental conditions to O on the Cu surface is needed to fully understand how subsurface O forms and this would be outside the scope of this present study.

Author Reply: Thank you for your helpful comments and suggestions. We are glad that you agree with our conclusions of the stabilities of subsurface and surface oxygen in Cu system, that is subsurface oxygen is stable in Cu system even it is less stable than surface oxygen. We agree with your opinion that the formation of subsurface in Cu system under experimental conditions is an interesting topic, but it is outside the scope of this present study. We will include this study in our future plan.

We in particular appreciate your strong support for publication in Nature Communications.